# A novel variational form of the Schatten-$p$ quasi-norm

**Paris Giampouras**
Mathematical Institute for Data Science
Johns Hopkins University
parisg@jhu.edu

**René Vidal**
Mathematical Institute for Data Science
Johns Hopkins University
rvidal@jhu.edu

**Athanasios Rontogiannis**
IAASARS
National Observatory of Athens
tronto@noa.gr

**Benjamin D. Haeffele**
Mathematical Institute for Data Science
Johns Hopkins University
bhaeffele@jhu.edu

## Abstract

The Schatten-$p$ quasi-norm with $p \in (0,1)$ has recently gained considerable attention in various low-rank matrix estimation problems offering significant benefits over relevant convex heuristics such as the nuclear norm. However, due to the nonconvexity of the Schatten-$p$ quasi-norm, minimization suffers from two major drawbacks: 1) the lack of theoretical guarantees and 2) the high computational cost which is demanded for the minimization task even for trivial tasks such as finding stationary points. In an attempt to reduce the high computational cost induced by Schatten-$p$ quasi-norm minimization, variational forms, which are defined over smaller-size matrix factors whose product equals the original matrix, have been proposed. Here, we propose and analyze a novel *variational form of Schatten-$p$ quasi-norm* which, for the first time in the literature, is defined for any continuous value of $p \in (0,1]$ and decouples along the columns of the factorized matrices. The proposed form can be considered as the natural generalization of the well-known variational form of the nuclear norm to the nonconvex case i.e., for $p \in (0,1)$. Notably, low-rankness is now imposed via a group-sparsity promoting regularizer. The resulting formulation gives way to SVD-free algorithms thus offering lower computational complexity than the one that is induced by the original definition of the Schatten-$p$ quasi-norm. A local optimality analysis is provided which shows that we can arrive at a local minimum of the original Schatten-$p$ quasi-norm problem by reaching a local minimum of the matrix factorization based surrogate problem. In addition, for the case of the squared Frobenius loss with linear operators obeying the restricted isometry property (RIP), a rank-one update scheme is proposed, which offers a way to escape poor local minima. Finally, the efficiency of our approach is empirically shown on a matrix completion problem.

## 1 Introduction

Recently, nonconvex heuristics such as $\ell_p$ and $\mathcal{S}_p$ (Schatten-$p$) quasi-norms[1] with $0 < p < 1$ have been shown to significantly outperform their convex counterparts, i.e., $\ell_1$ and nuclear norms, in sparse/low-rank vector/matrix recovery problems, for a wide range of applications [1–4]. Indeed, for applications such as recovering low-rank matrices from limited measurements the $\mathcal{S}_p$ quasi-norm for $p \in (0,1)$ is known theoretically to perform at least as well, if not better than the nuclear norm

(the closest convex approximation to the rank function), provided that one can find the minimum $\mathcal{S}_p$ solution [5]. This result is somewhat intuitive in the sense that it is known that the $\mathcal{S}_p$ quasi-norm provides an increasingly tight approximation to the rank of a matrix as $p \to 0$; however, the price to be paid is the fact that the $\mathcal{S}_p$ quasi-norm is no longer convex for $p \in (0, 1)$, which can significantly complicate efficiently solving models that employ the $\mathcal{S}_p$ quasi-norm.

Over the past few years, several works have been presented in the literature which aim to provide practical methods to approximately solve $\mathcal{S}_p$ quasi-norm minimization problems, [2, 3, 6–8]. By and large, these works typically consider problems of the form

$$\min_{\mathbf{X} \in \mathbb{R}^{m \times n}} l(\mathbf{Y}, \mathbf{X}) + \lambda \|\mathbf{X}\|_{\mathcal{S}_p}^p \tag{1}$$

where $\mathbf{Y}$ is a data matrix and $l(\mathbf{Y}, \mathbf{X})$ is a loss function which serves as a measure of how well the adopted model fits the data. Clearly, due to nonconvexity, the task of finding a minimizer of (1) can be quite challenging and potentially limits what can be guaranteed theoretically about solving problems with form (1). Moreover, traditional iterative $\mathcal{S}_p$ quasi-norm minimization algorithms typically rely on singular value decomposition steps (SVDs) at each iteration, whose computational cost ($\mathcal{O}(\min(mn^2, m^2n))$ ) may be cumbersome in the high dimensional and large-scale data regime.

With the aim to address the high computational cost of $\mathcal{S}_p$ quasi-norm minimization, variational forms of the $\mathcal{S}_p$ quasi-norm have been proposed in the literature. Such approaches first: 1) represent a given matrix $\mathbf{X} \in \mathbb{R}^{m \times n}$ as the product of matrices $\mathbf{U} \in \mathbb{R}^{m \times d}$ and $\mathbf{V} \in \mathbb{R}^{n \times d}$, i.e., $\mathbf{X} = \mathbf{U}\mathbf{V}^T$ and then 2) define a regularization term over $\mathbf{U}$ and $\mathbf{V}$ whose minimum equals the value of the $\mathcal{S}_p$ quasi-norm. For example, when $p = 1$, the $\mathcal{S}_p$ quasi-norm reduces to the nuclear norm $\|\mathbf{X}\|_* = \sum_{i=1}^{\min(m,n)} \sigma_i(\mathbf{X})$, for which several variational forms have been proposed in the literature [9]:

$$\|\mathbf{X}\|_* = \min_{d \in \mathbb{N}_+} \min_{\substack{\mathbf{U}, \mathbf{V} \\ \mathbf{U}\mathbf{V}^\top = \mathbf{X}}} \sum_{i=1}^{d} \|\mathbf{u}_i\|_2 \|\mathbf{v}_i\|_2 = \min_{d \in \mathbb{N}_+} \min_{\substack{\mathbf{U}, \mathbf{V} \\ \mathbf{U}\mathbf{V}^\top = \mathbf{X}}} \sum_{i=1}^{d} \tfrac{1}{2}(\|\mathbf{u}_i\|_2^2 + \|\mathbf{v}_i\|_2^2). \tag{2}$$

Then, instead of working with a problem with form (1) one can instead consider an equivalent[2] problem in the $(\mathbf{U}, \mathbf{V})$ space:

$$\min_{d \in \mathbb{N}_+} \min_{\mathbf{U}, \mathbf{V}} l(\mathbf{Y}, \mathbf{U}\mathbf{V}^T) + \lambda \sum_{i=1}^{d} \|\mathbf{u}_i\|_2 \|\mathbf{v}_i\|_2 = \min_{d \in \mathbb{N}_+} \min_{\mathbf{U}, \mathbf{V}} l(\mathbf{Y}, \mathbf{U}\mathbf{V}^T) + \lambda \sum_{i=1}^{d} \tfrac{1}{2}(\|\mathbf{u}_i\|_2^2 + \|\mathbf{v}_i\|_2^2) \tag{3}$$

Notably, although variational forms allow one to define problems in terms of $(\mathbf{U}, \mathbf{V})$ instead of $\mathbf{X}$, which potentially results in significant computational advantages for large-scale data, the resulting problems are in general nonconvex w.r.t. $\mathbf{U}$ and $\mathbf{V}$ by construction due to the presence of the matrix product $\mathbf{U}\mathbf{V}^\top$. However, despite this challenge, for the case of the convex nuclear norm ($p = 1$) global optimality guarantees can still be obtained for solving problems in the $(\mathbf{U}, \mathbf{V})$ domain, such as (3), by exploiting properties of the variational form [10].

Capitalizing on the merits of the variational form of the nuclear norm, different alternative definitions of the $\mathcal{S}_p$ quasi-norm have been proposed. In [3], a variational form of $\mathcal{S}_p$ is defined for $p = \{\frac{1}{2}, \frac{2}{3}\}$ via norms defined over the matrix factors $\mathbf{U}$ and $\mathbf{V}$. Along the same lines, a generalized framework which is valid for any value of $p$ was proposed in [11]; however, these forms typically require one to compute SVDs of the $(\mathbf{U}, \mathbf{V})$ factors to evaluate the (quasi-)norm on $\mathbf{X}$, which can be quite limiting computationally in the large-data setting. Recently, a new variational form of the $\mathcal{S}_p$ quasi-norm was introduced in [4], which, contrary to previous approaches of [11, 12], is defined over the columns of the matrix factors via group-sparsity imposing norms and are thus SVD-free. Nevertheless, a major shortcoming of the variational form proposed in [4] is that it is only defined for discrete values of $p$ rather than for an arbitrary $p \in (0, 1)$. As a consequence, it is not a full generalization of the variational form of the nuclear norm. Finally, all existing approaches for defining variational forms of the $\mathcal{S}_p$ quasi-norm lack any optimality analysis or study with regard to optimizing the problem in the resulting $(\mathbf{U}, \mathbf{V})$ space, and a significant question is whether such forms introduce additional 'spurious' local minima into the optimization landscape (i.e., a local minimum might exist for $(\mathbf{U}, \mathbf{V})$, but there is no local minimum if one is optimizing w.r.t. $\mathbf{X}$ at $\mathbf{X} = \mathbf{U}\mathbf{V}^\top$).

**Paper contributions**. In this paper, we propose a novel variational form of the $\mathcal{S}_p$ quasi-norm which, similarly to [4], is defined over the columns of the matrix factors $\mathbf{U}$ and $\mathbf{V}$, but unlike [4] our

proposed form is valid *for any continuous value of $p \in (0, 1)$*. Further, the proposed variational form of the $\mathcal{S}_p$ quasi-norm is a *direct generalization of the celebrated variational form of the nuclear norm* (2) to the nonconvex case i.e., for $p \in (0, 1)$, with a clear and intuitive link showing the generalization. In addition, by minimizing the proposed form, the resulting algorithms induce column sparsity on the factors after only a few iterations, eventually converging to matrix factors whose number of nonzero columns equals to their rank. In that sense, a *rank revealing* decomposition of the matrix $\mathbf{X}$ is obtained directly from the optimization w.r.t. $(\mathbf{U}, \mathbf{V})$.

Our second contribution is to present a theoretical analysis which shows that *local minimizers of the factorized problem give rise to local minimizers of the original problem defined w.r.t. to* $\mathbf{X}$. Our optimality analysis makes use of nonconvex optimization theory [13] and is general in the sense that it can potentially be applied to other variational forms of the $\mathcal{S}_p$ quasi-norm as well as to general concave singular value penalty functions on the condition that such functions are *subdifferentially regular* (see supplement). As a result of our analysis, one has the computational benefits imparted by the variational form with assurances that no additional poor local minima are introduced into the problem.

Our third contribution is to provide a strategy for escaping from "bad" stationary points by making use of rank-one updates for the case of the squared loss functions composed with linear operators that satisfy the restricted isometry property (RIP). In particular, when optimizing w.r.t. $(\mathbf{U}, \mathbf{V})$ one must choose an initialization for the rank (i.e., the number of columns in $\mathbf{U}$ and $\mathbf{V}$), and with a poor initial choice (e.g., initializing with too small of an initial rank) can result in convergence to poor local minima, which can be escaped by our rank-one update strategy by growing the rank of the solution.

In summary, the current paper goes beyond state-of-the-art by a) generalizing the ubiquitous varational form of the nuclear norm ($\mathcal{S}_p$ for $p = 1$) to the nonconvex case $p \in (0, 1)$ and b) providing a fruitful insight on the implications arising by transforming the original nonconvex problem into the factorized space vis-a-vis the landscape properties of the newly formulated nonconvex objective function.

## 2  Prior Art

The variational definition of the nuclear norm ($\mathcal{S}_p$ for $p = 1$) given in (2), has been widely utilized in a variety of problems. For example, formulations similar that in (3) have been employed for problems such as collaborative filtering [9], robust PCA [14], etc. In a similar vein, a few efforts for providing alternative formulations of the $\mathcal{S}_p$ quasi-norms for $p \in (0, 1)$ have recently appeared in the literature. In [12], the authors proposed variational definitions for the $\mathcal{S}_{\frac{1}{2}}$ quasi-norm i.e.,

$$\|\mathbf{X}\|_{\mathcal{S}_{\frac{1}{2}}}^{\frac{1}{2}} = \min_{d \in \mathbb{N}_+} \min_{\mathbf{U}\mathbf{V}^T = \mathbf{X}} \|\mathbf{U}\|_*^{\frac{1}{2}} \|\mathbf{V}\|_*^{\frac{1}{2}} = \min_{\mathbf{U}\mathbf{V}^\top = \mathbf{X}} \frac{\|\mathbf{U}\|_* + \|\mathbf{V}\|_*}{2}. \tag{4}$$

In [11], generalized variational forms of the $\mathcal{S}_p$ quasi-norm for any $p \in (0, 1)$ such that $\frac{1}{p} = \frac{1}{p_1} + \frac{1}{p_2}$ were derived:

$$\|\mathbf{X}\|_{\mathcal{S}_p}^p = \min_{d \in \mathbb{N}_+} \min_{\mathbf{U}\mathbf{V}^T = \mathbf{X}} \|\mathbf{U}\|_{\mathcal{S}_{p_1}}^p \|\mathbf{V}\|_{\mathcal{S}_{p_2}}^p = \min_{\mathbf{U}\mathbf{V}^T = \mathbf{X}} \frac{p_2 \|\mathbf{U}\|_{\mathcal{S}_{p_1}}^{p_1} + p_1 \|\mathbf{V}\|_{\mathcal{S}_{p_2}}^{p_2}}{p_1 + p_2}. \tag{5}$$

While the above variational form is general for an arbitrary value of $p \in (0, 1)$, note that in the general case one still needs to compute SVDs of $\mathbf{U}$ and $\mathbf{V}$ to evaluate the variational form due to the $\mathcal{S}_p$ quasi-norms on the factors, which can be a hindrance in large-scale problems. To alleviate this problem, the authors of [4] recently proposed two different variational definitions of the $\mathcal{S}_p$ quasi-norm defined for discrete values of $p$, $p = \frac{q}{q+1}$ and $p' = \frac{2q}{q+2}$ with $q = \{1, \frac{1}{2}, \frac{1}{4}, \ldots, \}$, as

$$\|\mathbf{X}\|_{\mathcal{S}_p}^p = \min_{d \in \mathbb{N}_+} \min_{\mathbf{U}\mathbf{V}^\top = \mathbf{X}} \frac{1}{(1 + \frac{1}{q})\alpha^{\frac{q}{q+1}}} \sum_{i=1}^{d} \frac{1}{q} \|\mathbf{u}_i\|_2^q + \alpha \|\mathbf{v}_i\|_2 \tag{6}$$

and

$$\|\mathbf{X}\|_{\mathcal{S}_{p'}}^{p'} = \min_{d \in \mathbb{N}_+} \min_{\mathbf{U}\mathbf{V}^\top = \mathbf{X}} \frac{1}{(\frac{1}{2} + \frac{1}{q})\alpha^{\frac{q}{q+2}}} \sum_{i=1}^{d} \frac{1}{q} \|\mathbf{u}_i\|_2^q + \frac{\alpha}{2} \|\mathbf{v}_i\|_2^2. \tag{7}$$

It should be noted that both of the above $\mathcal{S}_p$ quasi-norms are defined over the columns of the factors $\mathbf{U}$ and $\mathbf{V}$. Thus they provide potentially much simpler forms to evaluate in the $(\mathbf{U}, \mathbf{V})$ domain due to the fact that one does not need to compute any SVDs. Further, the forms in (6) and (7) can also be interpreted as providing a *group-sparsity* regularization on the columns of $(\mathbf{U}, \mathbf{V})$, which sets entire columns of $(\mathbf{U}, \mathbf{V})$ to 0. Despite these advantages, we note that existing forms are not natural generalizations of the classical result for the nuclear norm (2) and they are not valid for all continuous values $p \in (0, 1]$. Moreover, they have been proposed without a theoretical analysis of local or global optimality. Next, we propose an alternative variational form that addresses all of these shortcomings.

## 3 The variational form of the Schatten-$p$ quasi-norm

In this section, we introduce a variational form of the $\mathcal{S}_p$ (Schatten-$p$) quasi-norm which is defined for any continuous value of $p \in (0, 1)$. For its derivation, we invoke simple arguments of classical linear algebra for the properties of concave sums of singular values that date back to the 1970s [15].

**Theorem 1.** *(Theorem 3, [15]) Let $\mathbf{A}, \mathbf{B}, \mathbf{C}$ be $m \times n$ matrices with $m \geq n$ such that $\mathbf{C} = \mathbf{A} + \mathbf{B}$. For any permutation invariant, non-decreasing and concave function $F : \mathbb{R}^{2n}_+ \cup \mathbf{0} \to \mathbb{R}$, it holds*

$$F(c_1, c_2, \ldots, c_n, 0, 0, \ldots, 0) \leq F(a_1, a_2, \ldots, a_n, b_1, b_2, \ldots, b_n) \tag{8}$$

*where $a_i, b_i, c_i$ for $i = 1, 2, \ldots, n$ denote the singular values of matrices $\mathbf{A}, \mathbf{B}, \mathbf{C}$.*

The proposed variational form can be immediately derived from Theorem 1 as follows.

**Theorem 2.** *Let $0 < p \leq 1$, $\mathbf{X} \in \mathbb{R}^{m \times n}$ and $\mathrm{rank}(\mathbf{X}) = r$ where $r \leq d \leq \min(m, n)$. Then:*

$$
\begin{aligned}
\|\mathbf{X}\|^p_{\mathcal{S}_p} &= \min_{d \in \mathbb{N}_+} \min_{\mathbf{U} \in \mathbb{R}^{m \times d}, \mathbf{V} \in \mathbb{R}^{n \times d}} \sum_{i=1}^{d} \|\mathbf{u}_i\|^p_2 \|\mathbf{v}_i\|^p_2 \quad \text{s.t. } \mathbf{U}\mathbf{V}^\top = \mathbf{X} \\
&= \min_{d \in \mathbb{N}_+} \min_{\mathbf{U} \in \mathbb{R}^{m \times d}, \mathbf{V} \in \mathbb{R}^{n \times d}} \frac{1}{2^p} \sum_{i=1}^{d} (\|\mathbf{u}_i\|^2_2 + \|\mathbf{v}_i\|^2_2)^p \quad \text{s.t. } \mathbf{U}\mathbf{V}^\top = \mathbf{X}
\end{aligned}
\tag{9}
$$

*Proof.* Let us define $F : \mathbb{R}^{2n}_+ \to \mathbb{R}$ as $F(x_1, x_2, \ldots, x_{2n}) = \sum_{i=1}^{2n} f(x_i)$ where $f(x_i) = |x_i|^p$ with $p \leq 1$. Clearly, $F$ satisfies the conditions of Theorem 1, i.e., $F$ is permutation invariant, non-decreasing and concave. Now, since $\mathrm{rank}(\mathbf{X}) \leq d$, we can express $\mathbf{X}$ as the product of matrices $\mathbf{U} \in \mathbb{R}^{m \times d}$ and $\mathbf{V} \in \mathbb{R}^{n \times d}$, i.e., $\mathbf{X} = \mathbf{U}\mathbf{V}^T = \sum_{i=1}^{d} \mathbf{u}_i \mathbf{v}_i^T$, where $d \geq r$. Let us define $\mathbf{A} = \mathbf{u}_1 \mathbf{v}_1^\top$ and $\mathbf{B} = \sum_{i=2}^{d} \mathbf{u}_i \mathbf{v}_i^\top$. Matrices $\mathbf{u}_i \mathbf{v}_i^T$ and $\mathbf{B}$ consist of no more than 1 and $d - 1$ non-zero singular values, respectively. From Theorem 1, we get $F(\sigma_1(\mathbf{X}), \sigma_2(\mathbf{X}), \ldots, \sigma_r(\mathbf{X}), 0, \ldots, 0_{2n-r}) \leq F(\sigma_1(\mathbf{u}_1 \mathbf{v}_1^\top), 0, \ldots, 0_{n-1}, \sigma_1(\mathbf{B}), \sigma_2(\mathbf{B}), \ldots, \sigma_{d-1}(\mathbf{B}), 0, \ldots, 0_{n-(d-1)})$, which can be written as

$$\sum_{i=1}^{r} \sigma_i^p(\mathbf{X}) \leq \sigma_1^p(\mathbf{u}_1 \mathbf{v}_1^\top) + \sum_{i=1}^{d-1} \sigma_i^p(\mathbf{B}). \tag{10}$$

Applying the same decomposition step followed for $\mathbf{X}$ to $\mathbf{B}$ and repeating the process for $d - 2$ steps leads to the following inequality:

$$\sum_{i=1}^{r} \sigma_i^p(\mathbf{X}) \leq \sum_{i=1}^{d} \sigma_1^p(\mathbf{u}_i \mathbf{v}_i^\top). \tag{11}$$

Then since $\sigma_1(\mathbf{u}_i \mathbf{v}_i^T) = \|\mathbf{u}_i \mathbf{v}_i^T\|_F = \|\mathbf{u}_i\|_2 \|\mathbf{v}_i\|_2$ and $\|\mathbf{u}_i\|_2 \|\mathbf{v}_i\|_2 \leq \frac{1}{2}(\|\mathbf{u}_i\|^2_2 + \|\mathbf{v}_i\|^2_2)$, we obtain the two forms appearing in the RHS of (9). To show that the inequality is actually achieved, and thus the minimum of the RHS of (9) is equal to the LHS, observe that the minimum is always achieved for $\mathbf{U} = \mathbf{U}_{\mathbf{X}} \boldsymbol{\Sigma}^{\frac{1}{2}}$ and $\mathbf{V} = \mathbf{V}_{\mathbf{X}} \boldsymbol{\Sigma}^{\frac{1}{2}}$, where $\mathbf{X} = \mathbf{U}_{\mathbf{X}} \boldsymbol{\Sigma} \mathbf{V}_{\mathbf{X}}^T$ is the singular value decomposition of $\mathbf{X}$. $\square$

It should be noted that a general result of the same type as ours was developed recently appeared in [16] for concave singular value functions. The proof of [16] is also simple, however it was based on a more complicated result (Theorem 4.4. of [17], instead of the simpler Theorem 1 of [15] applied in our case. Surprisingly, classical linear algebra results from [15, 17] appear to have been overlooked

when deriving variational forms of the $\mathcal{S}_p$ quasi-norm in the machine learning literature, leading to more complicated derivations of alternative variational forms of the $\mathcal{S}_p$ quasi-norm, such as (6) and (7) in [4], which are defined only for discrete values of $p$. In contrast, our proposed variational form can be defined for any continuous value of $p \in (0, 1)$. In addition, our proposed variational form is a natural generalization of the variational form of the nuclear norm for $p = 1$ given in (2). Moreover, unlike the forms in (4) and (5), our form depends only on the columns of $\mathbf{U}$ and $\mathbf{V}$, which will allow more efficient optimization methods (which do not require SVD calculations at each optimization iteration) for matrix factorization problems with $\mathcal{S}_p$ quasi-norm regularization, such as (1), by restating them w.r.t the matrix factors $\mathbf{U}, \mathbf{V}$ as follows[3]:

$$
\begin{aligned}
\min_{\mathbf{X}} l(\mathbf{Y}, \mathbf{X}) + \lambda\|\mathbf{X}\|_{\mathcal{S}_p}^p &= \min_{\mathbf{U}, \mathbf{V}} l(\mathbf{Y}, \mathbf{U}\mathbf{V}^T) + \lambda \sum_{i=1}^{d} \|\mathbf{u}_i\|_2^p \|\mathbf{v}_i\|_2^p \\
&= \min_{\mathbf{U}, \mathbf{V}} l(\mathbf{Y}, \mathbf{U}\mathbf{V}^T) + \frac{\lambda}{2^p} \sum_{i=1}^{d} (\|\mathbf{u}_i\|_2^2 + \|\mathbf{v}_i\|_2^2)^p.
\end{aligned}
\tag{12}
$$

# 4 Analysis of the Landscape of the Non-Convex Objective

In this section, we present an analysis of the problem defined in (12) elaborating on the conditions that ensure that a local minimum of the variationally defined $\mathcal{S}_p$ quasi-norm minimization problem, i.e., a pair $(\hat{\mathbf{U}}, \hat{\mathbf{V}})$, gives rise to a matrix $\hat{\mathbf{X}} = \hat{\mathbf{U}}\hat{\mathbf{V}}^T$ which is a local minimum of the original $\mathcal{S}_p$ quasi-norm regularized objective function defined in (1). This is motivated by the fact that when one changes the domain from $\mathbf{X}$ to $(\mathbf{U}, \mathbf{V})$ then it is possible to introduce additional 'spurious' local minima into the problem, i.e., a point which is local minima w.r.t. $(\mathbf{U}, \mathbf{V})$ but such that $\mathbf{X} = \mathbf{U}\mathbf{V}^\top$ is not a local minima w.r.t. $\mathbf{X}$. It has been shown that in certain settings, such as semidefinite programming problems in standard form [18], solving problems in the factorized domain will not introduce additional local minima, but as problems involving the $\mathcal{S}_p$ quasi-norm with $p \in (0, 1)$ are inherently non-convex even in the $\mathbf{X}$ domain, it is unclear whether such results can be generalized to this setting. Here we provide a positive result and show that under mild conditions local minima w.r.t. $(\mathbf{U}, \mathbf{V})$ will correspond to local minima w.r.t. $\mathbf{X}$. In addition, we also consider a special case of the general problem in (1) where the loss is chosen to be the squared Frobenius composed with RIP linear operators. In this setting we propose a rank-one update strategy to escape poor stationary points and local minima by growing the rank of the factorization.

## 4.1 Properties of Local Minima in Factorized Schatten-$p$ Norm Regularized Problems

The general idea that motivates this section is similar to what has been used to study similar variational forms defined for other general norms on $\mathbf{X}$ [10, 19]. Concretely, the key approach employed in this prior work is to relate the local minima of *nonconvex* objective functions that are defined w.r.t. matrices $\mathbf{U}$ and $\mathbf{V}$ to global minima of *convex* objective functions over $\mathbf{X}$ such that $\mathbf{X} = \mathbf{U}\mathbf{V}^T$. However, it is worth emphasizing that since in our case we focus on $\mathcal{S}_p$ quasi-norms for values of $p \in (0, 1)$, both the problems w.r.t. to $\mathbf{X}$ and $\mathbf{U}, \mathbf{V}$ are nonconvex. Evidently, this presents considerable challenges in deriving conditions for global optimality w.r.t. either $\mathbf{X}$ or $(\mathbf{U}, \mathbf{V})$. Moreover, finding even a local minimum of a nonconvex problem can be NP-hard in general. Interestingly, as we show in Theorem 3, when certain conditions are satisfied we ensure that no decreasing directions of the objective function w.r.t. $\mathbf{X}$ exist, thus showing that we have arrived at a local minimum of (12) and (1).

The first step in establishing optimality properties of the $\mathcal{S}_p$ quasi-norm regularized problem given in (1) is to derive the subgradients of the $\mathcal{S}_p$ quasi-norm. Note that conventional definitions of subgradients are not valid in the case of the $\mathcal{S}_p$ quasi-norm due to the lack of convexity. That said, generalized notions of subgradients (referred to as *regular subgradients*, [13]) can be utilized to account for local variations of the objective functions. In the following Lemma, we derive the regular subdifferential of the $\mathcal{S}_p$ quasi-norm.

**Lemma 1.** *Let* $\mathbf{X} = \mathbf{U}\boldsymbol{\Sigma}\mathbf{V}^T$ *where* $\mathbf{U} \in \mathbb{R}^{m \times r}, \boldsymbol{\Sigma} \in \mathbb{R}^{r \times r}$ *and* $\mathbf{V} \in \mathbb{R}^{n \times r}$, *denote the singular value decomposition of* $\mathbf{X} \in \mathbb{R}^{m \times n}$. *The regular subdifferential of* $\|\mathbf{X}\|^p_{\mathcal{S}_p}$ *is given by*

$$\hat{\partial}\|\mathbf{X}\|^p_{\mathcal{S}_p} = \{\mathbf{U}\mathrm{diag}(\hat{\partial}\|(\boldsymbol{\sigma}_+(\mathbf{X})\|^p_p)\mathbf{V}^T + \mathbf{W} : \mathbf{U}^T\mathbf{W} = \mathbf{0}, \mathbf{W}\mathbf{V} = \mathbf{0}\} \tag{13}$$

*where* $\boldsymbol{\sigma}_+(\mathbf{X})$ *is the vector of the positive singular values of* $\mathbf{X}$.

Note that the form of the expressions derived for regular subgradients of Schatten-$p$ quasi-norm resemble the ones of subgradients of the nuclear norm, however the former a) are locally defined and b) require no constraint on the spectral norm of $\mathbf{W}$.

Next we relate local minima of the factorized problem defined over $\mathbf{U}, \mathbf{V}$ to local minima of the original $\mathcal{S}_p$ quasi-norm problem defined w.r.t. $\mathbf{X}$ in (1).

**Theorem 3.** *Let* $(\hat{\mathbf{U}}, \hat{\mathbf{V}})$ *be a local minimizer of the factorized* $\mathcal{S}_p$ *quasi-norm regularized objective functions defined in* (12), *where* $\hat{\mathbf{U}} = \mathbf{U}\boldsymbol{\Sigma}^{\frac{1}{2}}$ *and* $\hat{\mathbf{V}} = \mathbf{V}\boldsymbol{\Sigma}^{\frac{1}{2}}$, *matrices* $\mathbf{U} \in \mathbb{R}^{m \times m}$ *and* $\mathbf{V} \in \mathbb{R}^{n \times m}$, *consisting of* $r < m \leq n$ *nonzero orthonormal columns, and* $\boldsymbol{\Sigma}$ *is a* $m \times m$ *real-valued diagonal matrix. Assume matrix* $\hat{\mathbf{X}} = \hat{\mathbf{U}}\hat{\mathbf{V}}^T = \mathbf{U}\boldsymbol{\Sigma}\mathbf{V}^T$ *with* $\mathrm{rank}(\hat{\mathbf{X}}) \leq r$. $\hat{\mathbf{X}}$ *is a local minimizer of the* $\mathcal{S}_p$ *regularized objective function defined over* $\mathbf{X}$ *in (1).*

Theorem 3 in fact says that we can always obtain a local minimizer of the original objective function defined over $\mathbf{X}$ once we reach a pair $(\hat{\mathbf{U}}, \hat{\mathbf{V}})$, which is a local minimizer of the factorized $\mathcal{S}_p$ regularized objective function, on the condition that it gives rise to a low-rank matrix $\hat{\mathbf{X}}$ (which is akin to requiring that $(\mathbf{U}, \mathbf{V})$ is sufficiently parameterized). While the above result also requires the nonzero columns of $(\hat{\mathbf{U}}, \hat{\mathbf{V}})$ to be orthogonal, we note that this is not a significant limitation in practice as one can always perform a single SVD of $\hat{\mathbf{X}} = \hat{\mathbf{U}}\hat{\mathbf{V}}^\top$ to find a factorization with the required form which has objective value in (12) less than or equal to that of $(\hat{\mathbf{U}}, \hat{\mathbf{V}})$.

### 4.2 Rank One Updates for Escaping Poor Local Minima

Next we consider the special case where the loss function is the squared loss composed with a linear operator (here we show one of the two variational forms that we propose, but everything in this section applies equivalently to the other form), which gives the following form:

$$\min_{\mathbf{U},\mathbf{V}} \tfrac{1}{2}\|\mathbf{Y} - \mathcal{A}(\mathbf{U}\mathbf{V}^\top)\|^2_F + \frac{\lambda}{2^p}\sum_{i=1}^d \left(\|\mathbf{u}_i\|^2_2 + \|\mathbf{v}_i\|^2_2\right)^p, \tag{14}$$

where $\mathcal{A}$ is a general linear operator. We are then interested in whether a rank-one descent step can be found after converging to a stationary point or local minima. That is, can we augment our $(\mathbf{U}, \mathbf{V})$ factors by adding an additional column which will improve the objective function and allow us to escape from poor local minima or stationary points. In particular, we want to solve the following:

$$\min_{\bar{\mathbf{u}},\bar{\mathbf{v}}} \tfrac{1}{2}\|\mathbf{Y} - \mathcal{A}(\mathbf{U}\mathbf{V}^\top + \bar{\mathbf{u}}\bar{\mathbf{v}}^\top)\|^2_F + \lambda\sum_{i=1}^d \left(\frac{\|\mathbf{u}_i\|^2_2 + \|\mathbf{v}_i\|^2_2}{2}\right)^p + \lambda\left(\frac{\|\bar{\mathbf{u}}\|^2_2 + \|\bar{\mathbf{v}}\|^2_2}{2}\right)^p. \tag{15}$$

Note that in the $p = 1$ case this is the variational form of a nuclear norm problem, and if the optimal $(\bar{\mathbf{u}}, \bar{\mathbf{v}})$ is the all zero vectors, then this is sufficient to guarantee the global optimality of $(\mathbf{U}, \mathbf{V})$ for the problem in (14) with $p = 1$ [10]. Further, for the $p = 1$ case it can be shown that the solution to (15) can be solved via a singular vector problem [10]. Namely, the optimal $(\bar{\mathbf{u}}, \bar{\mathbf{v}})$ will be a scaled multiple of the largest singular vector pair of the matrix $\mathcal{A}^*(\mathbf{Y} - \mathcal{A}(\mathbf{U}\mathbf{V}^\top))$, where $\mathcal{A}^*$ denotes the adjoint operator of $\mathcal{A}$.

For the $p < 1$ case solving (15) may not be sufficient to guarantee global optimality, but this can still provide a practical solution for improving a given stationary point solution. One difficulty, however, is that there is always a local minima around the origin for problem (15) w.r.t. $(\bar{\mathbf{u}}, \bar{\mathbf{v}})$ [20] (i.e., for any $(\bar{\mathbf{u}}, \bar{\mathbf{v}})$ with sufficiently small magnitude the objective in (15) will be greater than for $(\bar{\mathbf{u}}, \bar{\mathbf{v}}) = (0, 0)$). As a result, we need to test if for a given direction $(\bar{\mathbf{u}}, \bar{\mathbf{v}})$ there exists a (arbitrarily large) scaling of $(\bar{\mathbf{u}}, \bar{\mathbf{v}})$ which reduces the objective. Specifically, note that (15) is equivalent to solving the following:

$$
\begin{aligned}
&\arg\min_{\tau \geq 0, \mathbf{u}, \mathbf{v}} \tfrac{1}{2}\|\mathbf{Y} - \mathcal{A}(\mathbf{U}\mathbf{V}^\top + \tau^2\mathbf{u}\mathbf{v}^\top)\|^2_F + \lambda\tau^{2p} \text{ s.t. } \|\mathbf{u}\|_2 = 1, \|\mathbf{v}\|_2 = 1\\
&= \arg\min_{\tau \geq 0, \mathbf{u}, \mathbf{v}} -\tau^2\langle\mathcal{A}^*(\mathbf{R}), \mathbf{u}\mathbf{v}^\top\rangle + \tfrac{1}{2}\tau^4\|\mathcal{A}(\mathbf{u}\mathbf{v}^\top)\|^2_F + \lambda\tau^{2p} \text{ s.t. } \|\mathbf{u}\|_2 = 1, \|\mathbf{v}\|_2 = 1
\end{aligned}
\tag{16}
$$

where $\mathbf{R} = \mathbf{Y} - \mathcal{A}(\mathbf{U}\mathbf{V}^\top)$. If the optimal solution for $\tau$ is at 0, then there is no rank-1 update that can decrease the objective, while if the optimal $\tau > 0$ then a rank-1 update can reduce the objective. Here we will consider the special case when the linear operator, $\mathcal{A}$, satisfies the RIP condition for rank-1 matrices with constant $\delta$, which is defined as follows:

**Definition 1.** *Let $\mathcal{A} : \mathbb{R}^{m \times n} \to \mathbb{R}^q$ be a linear operator and assume without loss of generality that $m \leq n$. We define the restricted isometry constant $0 \leq \delta < 1$ as the smallest number such that*

$$(1 - \delta)\|\mathbf{X}\|_F^2 \leq \|\mathcal{A}(\mathbf{X})\|^2 \leq (1 + \delta)\|\mathbf{X}\|_F^2 \tag{17}$$

*holds for any rank-one matrix $\mathbf{X}$, where $\|\mathcal{A}(\mathbf{X})\|$ denotes the spectral norm of $\mathcal{A}(\mathbf{X})$.*

From this, we then have the following rank-one update step:

**Proposition 1.** *For a linear operator $\mathcal{A}$ which is RIP on rank-1 matrices with constant $\delta = 0$ the solution to (16) $(\mathbf{u}_{opt}, \mathbf{v}_{opt})$ is given by the largest singular vector pair of $-\mathcal{A}^*(\mathbf{R})$. Further, let $\mu = \frac{2-2p}{2-p}\sigma(\mathcal{A}^*(\mathbf{R}))$, where $\sigma(\cdot)$ denotes the largest singular value. Then the optimal value of $\tau$ is given as:*

$$\tau_{opt} = \begin{cases} \sqrt{\mu} & \lambda - \mu^{1-p}\sigma(\mathcal{A}^*(\mathbf{R})) + \frac{1}{2}\mu^{2-p} \leq 0 \\ 0 & \text{otherwise} \end{cases} \tag{18}$$

*Additionally, for a linear operator $\mathcal{A}$ which is RIP on rank-1 matrices with constant $\delta > 0$, then the above solution will be optimal to within $\mathcal{O}(\frac{1}{2}\mu^2\delta)$ in objective value. Further, the above rank-1 update is guaranteed to reduce the objective provided $\lambda - \mu^{1-p}\sigma(\mathcal{A}^*(\mathbf{R})) + \frac{1}{2}\mu^{2-p} < -\frac{1}{2}\mu^2\delta$.*

## 5 Application to Schatten-$p$ Norm Regularized Matrix Completion

In this section we apply our framework to the matrix completion problem, where the goal is to estimate the missing entries of a data matrix by leveraging inherent low-rank structures in the data. We assume that we partially observe a noisy version of the true matrix $\mathbf{X}_{true}$ i.e.,

$$\mathcal{P}_Z(\mathbf{Y}) = \mathcal{P}_Z(\mathbf{X}_{true} + \mathbf{E}), \tag{19}$$

where $Z$ contains the indexes of the known elements of $\mathbf{Y}$, $\mathcal{P}_Z$ is a projection operator on this set and $\mathbf{E}$ contains zero-mean i.i.d. Gaussian elements with variance $\sigma^2$.

Assuming a low-rank representation of $\mathbf{X}_{true}$, we wish to find $\mathbf{X} = \mathbf{U}\mathbf{V}^T$ using our proposed variational form of the $\mathcal{S}_p$ quasi-norm by solving the following problem:

$$\min_{\mathbf{U},\mathbf{V}} \quad \frac{1}{2}\|\mathcal{P}_Z(\mathbf{Y} - \mathbf{U}\mathbf{V}^T)\|_F^2 + \frac{\lambda}{2^p}\sum_{i=1}^d (\|\mathbf{u}_i\|_2^2 + \|\mathbf{v}_i\|_2^2)^p. \tag{20}$$

The minimization task w.r.t. to matrix factors $\mathbf{U}, \mathbf{V}$ does not lead to closed form updates for $\mathbf{U}, \mathbf{V}$ due to both the form of the loss function and the non-separability of the regularization term for $p < 1$. That said, a block successive upper-bound minimization strategy is employed, which makes use of local tight upper-bounds of the objective function for updating each of the matrix factors [21]. The resulting algorithm does not entail singular value decomposition (SVD) steps and its computational complexity is in the order of $\mathcal{O}(|Z|d)$. Analytical details of the algorithm are provided in the supplement.

Recall also from Theorem 2 that the proposed objective in (20) is equivalent to a $\mathcal{S}_p$ quasi-norm regularized matrix completion formulation $\frac{1}{2}\|\mathcal{P}_Z(\mathbf{Y} - \mathbf{X})\|_F^2 + \lambda\|\mathbf{X}\|_{\mathcal{S}_p}^p$. Hence, generalization error bounds that have been provided for the $\mathcal{S}_p$ quasi-norm (Theorem 3 of [4]), which show that the matrix completion error can potentially be decreased as $p \to 0$, also apply here. The latter, is also empirically verified in the following subsection (see Figure 2)).

### 5.1 Experimental Results

In this section we provide simulated and real data experimental results that advocate the merits of the proposed variational form of the $\mathcal{S}_p$ quasi-norm in the case of the matrix completion problem. For comparison purposes, the iterative reweighted nuclear norm (IRNN) algorithm of [22], which makes use of a reweighted strategy for minimizing various concave penalty functions over the vector of singular values of the original matrix $\mathbf{X}$, is utilized. In our case, IRNN is set up for minimizing the

$\mathcal{S}_{\frac{1}{2}}$ quasi-norm. In addition, the softImpute-ALS algorithm of [23], which minimizes the variational form of the nuclear norm (see eq. (2)) and the factored group sparse (FGSR) algorithm for $\mathcal{S}_p$ quasi-norm minimization of [4], are also used as baseline methods. Note that the FGSR algorithm is based on the minimization of the variational $\mathcal{S}_p$ quasi-norm form given in (7) and hence is an algorithm which is closest to our approach. The regularization parameters for all algorithms are carefully selected so that they all attain their best performance. All experiments are conducted on a MacBook Pro with 2.6 GHz 6-Core Intel Core i7 CPU and 16GB RAM using MATLAB R2019b.

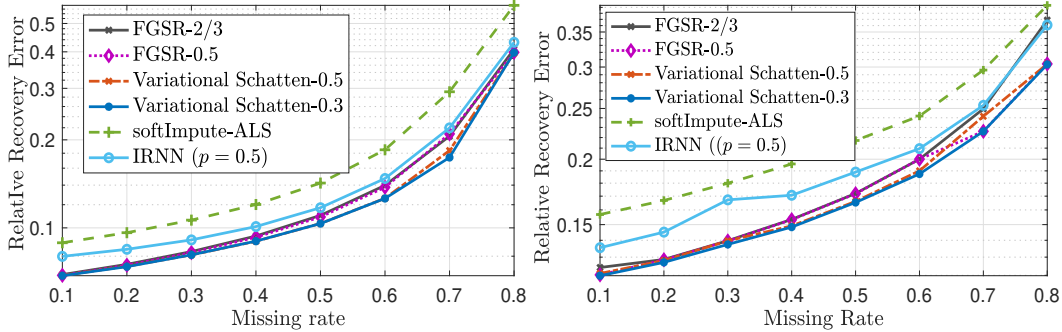

Figure 1: Relative Recovery Error vs Missing rate percentage for (a) SNR=15dB (left), true rank 50 and initial rank 75 (b) SNR=8dB (right), true rank 20 and initial rank 30.

**Simulated data**. We randomly generate a low-rank matrix of size $500 \times 500$ and rank $r$. The low-rank matrix is contaminated by additive i.i.d. Gaussian noise with $\sigma$ selected so that we get different SNR values for the input data matrix given to the algorithms. A subset of the elements of the noisy matrix is then selected uniformly at random. For each experiment, we report the average results of 10 independent runs. The relative recovery error is used as a performance metric defined as $\text{RE} = \frac{\|\mathcal{P}_{\bar{Z}}(\hat{\mathbf{X}} - \mathbf{X}_{true})\|_F}{\|\mathcal{P}_{\bar{Z}}(\mathbf{X}_{true})\|_F}$ (where $\bar{Z}$ contains the indices of the missing entries and $\hat{\mathbf{X}}$ denotes the estimated $\mathbf{X}$). In Figure 1, we see that the proposed algorithms (named Variational Schatten-$p$ with $p = \{0.3, 0.5\}$ in Figures 1(a) and 1(b)) perform equal or better than competing methods in virtually all situations. In all examined cases it can be seen that the

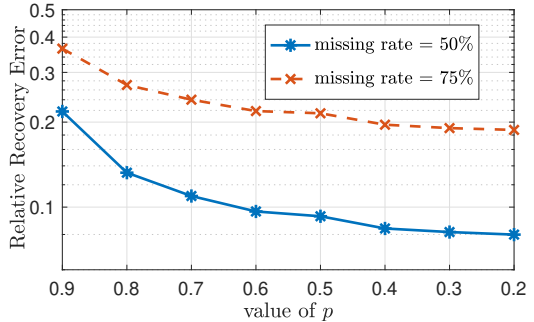

Figure 2: The effect of decreasing $p$ in the performance of the proposed variational $\mathcal{S}_p$ minimization algorithms.

softImpute-ALS, which uses the variational form of the nuclear norm shows poor performance as compared to the other algorithms that minimize either the original $\mathcal{S}_p$ quasi-norm or variational definitions thereof, highlighting the benefit of using the $\mathcal{S}_p$ quasi-norm with $p < 1$. In Table 1, we report the relative recovery error (average of 10 independent runs) as well as the (median) estimated final rank of the solution obtained when the algorithms are initialized with different rank initializations. As can be observed, the proposed algorithm is significantly more robust to errors in the rank of the initialization. Notably, the rank-one updating scheme detailed in Section 4.2, which allows the proposed algorithms to escape poor local minima, further enhances robustness to rank initialization by promoting its convergence to the true rank even in the more challenging scenario whereby the true rank is underestimated at initialization.

**Real data**. We next test the algorithms on the MovieLens-100K dataset, [24] which contains 100,000 ratings (integer values from 1 to 5) for 943 movies by 1682 users. We examine two cases corresponding to sampling rates of 50% and 25% of the known entries. For each case we initialize all matrix factorization based algorithms with ranks ranging from 10 to 50 with step-size 5. From Figure 3 it can be seen that the two versions of proposed algorithm corresponding to $p = 0.5$ and $p = 0.3$ performs comparably to FGSR-2/3 and FGSR-0.5 while outperforming IRNN and softImpute-ALS in terms of the normalized mean average error (NMAE). The latter, is shown to be vulnerable to

Table 1: Relative recovery error (RE) and (median) estimated rank $\hat{r}$ vs rank initialization. True rank $r = 20$, missing rate=0.4 and SNR=10dB.

| initial rank | Variational Schatten-0.5 | | | | Variational Schatten-0.3 | | | | FGSR-$2/3$ | | FGSR-0.5 | |
| | rank-one updates | | | | rank-one updates | | | | | | | |
| | yes | | no | | yes | | no | | | | | |
| | RE | $\hat{r}$ | RE | $\hat{r}$ | RE | $\hat{r}$ | RE | $\hat{r}$ | RE | $\hat{r}$ | RE | $\hat{r}$ |
|---|---|---|---|---|---|---|---|---|---|---|---|---|
| $0.5r$ | 0.2241 | 20 | 0.5260 | 10 | 0.2425 | 18 | 0.5263 | 10 | 0.5259 | 10 | 0.5259 | 10 |
| $0.75r$ | 0.1450 | 20 | 0.3253 | 10 | 0.1699 | 19 | 0.3254 | 10 | 0.3267 | 10 | 0.3253 | 10 |
| $r$ | 0.1228 | 20 | 0.1228 | 20 | 0.1225 | 20 | 0.1225 | 20 | 0.1320 | 20 | 0.1230 | 20 |
| $1.25r$ | 0.1231 | 20 | 0.1231 | 20 | 0.1219 | 20 | 0.1219 | 20 | 0.1320 | 20 | 0.1227 | 20 |
| $1.5r$ | 0.1231 | 20 | 0.1230 | 20 | 0.1223 | 20 | 0.1223 | 20 | 0.1325 | 20 | 0.1231 | 20 |

rank initialization unlike the remaining algorithms, which minimize different versions of the $\mathcal{S}_p$ quasi-norm.

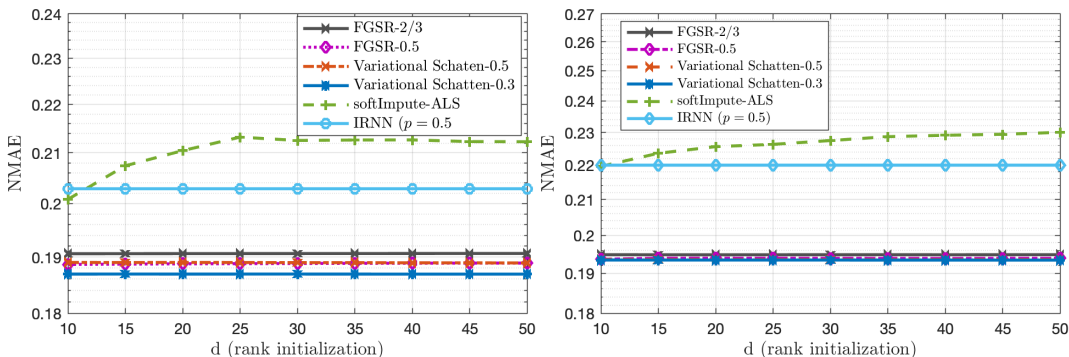

Figure 3: NMAE vs rank initialization for fraction of observed entries of 0.5 (left) and 0.25 (right).

# 6   Conclusions

In this work, a novel variational form of the Schatten-$p$ quasi-norm, which generalizes the popular variational form of the nuclear norm to the nonconvex case for $p \in (0, 1)$ is introduced. A local optimality analysis is provided, which shows how local minima of the variational problem correspond to local minima of the original one. A rank-one update scheme is given for the case of the Frobenius loss functions with RIP linear operators, which allows one to escape poor local minima. The merits of the proposed algorithm w.r.t. relevant state-of-the-art approaches are demonstrated on the matrix completion problem.

## Broader Impact

Low-rank modeling and estimation is a fundamental tool in machine learning and has numerous applications such as matrix completion and recommendation systems. As a result, understanding models for low-rank modeling and estimation is critical to understanding any potential biases and failure risks of such models. The proposed work offers new insights when it comes to the optimality properties of these problems, which might be of further interest to scientists studying relevant nonconvex theory. Moreover, our results provide theoretical insights and guidance which might be of interest to practitioners in guaranteeing and understanding the performance of their models.

## Acknowledgments and Disclosure of Funding

This work is partially supported by the European Union under the Horizon 2020 Marie-Skłodowska-Curie Global Fellowship program: HyPPOCRATES— H2020-MSCA-IF-2018, Grant Agreement Number: 844290, NSF Grants 2031985, 1934931, 1704458, and Northrop Grumman Research in Applications for Learning Machines (REALM) Initiative.

## Footnotes

[1]The $\mathcal{S}_p$ quasi-norm (raised to the power $p$) is defined as $\|\mathbf{X}\|_{\mathcal{S}_p}^p = \sum_{i=1} \sigma_i^p(\mathbf{X})$, where $\sigma_i(\mathbf{X})$ is the $i$th singular value of $\mathbf{X}$.

[2]Here equivalence is meant to imply that a solution for $(\mathbf{U}, \mathbf{V})$ also gives a solution for $\mathbf{X} = \mathbf{U}\mathbf{V}^\top$.

[3]Note that as in (2) all variational forms allow the number of columns in $(\mathbf{U}, \mathbf{V})$ to be arbitrary, but we will omit this for notational simplicity.

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
