[Supplementary Material]

# Supplementary Material of
# A novel variational form of the Schatten-$p$ quasi-norm

**Paris Giampouras**
Mathematical Institute for Data Science
Johns Hopkins University
parisg@jhu.edu

**René Vidal**
Mathematical Institute for Data Science
Johns Hopkins University
rvidal@jhu.edu

**Athanasios Rontogiannis**
IAASARS
National Observatory of Athens
tronto@noa.gr

**Benjamin D. Haeffele**
Mathematical Institute for Data Science
Johns Hopkins University
bhaeffele@jhu.edu

The $\mathcal{S}_p$ regularized objective function defined over $\mathbf{X}$ and the factorized objective functions defined over matrices $\mathbf{U}$ and $\mathbf{V}$ are given next,

$$\mathcal{L}(\mathbf{X}) = l(\mathbf{Y}, \mathbf{X}) + \lambda \|\mathbf{X}\|_{\mathcal{S}_p}^p \tag{1}$$

$$\mathcal{L}_1(\mathbf{U}, \mathbf{V}) = l(\mathbf{Y}, \mathbf{U}\mathbf{V}^\top) + \lambda \sum_{i=1}^{d} \|\mathbf{u}_i\|_2^p \|\mathbf{v}_i\|_2^p \tag{2}$$

$$\mathcal{L}_2(\mathbf{U}, \mathbf{V}) = l(\mathbf{Y}, \mathbf{U}\mathbf{V}^\top) + \frac{\lambda}{2^p} \sum_{i=1}^{d} (\|\mathbf{u}_i\|_2^2 + \|\mathbf{v}_i\|_2^2)^p. \tag{3}$$

## 1 Technical background and proofs of Lemma 1 and Theorem 3

First we provide definitions and the technical lemmas which are necessary for deriving: a) the regular subgradients of the Schatten-$p$ raised to $p$ quasi-norm for $p \in (0, 1)$ and b) the dual relationship between subderivatives and regular subgradients. Both are key ingredients of the proof of Theorem 3.

**Definition 2** ( [1]). *A function is called as a singular value function if it is extended real-valued, defined on $\mathbb{R}^{m \times n}$ of the form $f \circ \boldsymbol{\sigma} : \mathbb{R}^{m \times n} \to \mathbb{R}$, where $f : \mathbb{R}^q \to [-\infty, +\infty]$, $q \leq \min(m, n)$, is absolutely symmetric i.e., it is invariant to permutations and changes of the signs of its arguments.*

Based on Definition 1, we can say that Schatten-$p$ quasi-norm of matrix $\mathbf{X}$ is a singular value function arising from the $\ell_p$ quasi-norm, which is absolutely symmetric and is applied on the vector of the singular values of $\mathbf{X}$.

Next we provide the following notions of general, regular and horizon subgradients, which generalize traditional subgradients of convex functions to the case of nonconvex ones.

**Definition 3** ( [2]). *Let $f : \mathbb{R}^n \to \bar{\mathbb{R}}$ and a point $\bar{\mathbf{x}}$ with $f(\bar{\mathbf{x}})$ finite. A vector $\mathbf{u} \in \mathbb{R}^n$ is:*

- *a regular subgradient of $f$ at $\bar{x}$ i.e., $\mathbf{u} \in \hat{\partial} f(\mathbf{x})$, if*

$$f(\mathbf{x}) \geq f(\bar{\mathbf{x}}) + \langle \mathbf{u}, \mathbf{x} - \bar{\mathbf{x}} \rangle + \mathcal{O}(\|\mathbf{x} - \bar{\mathbf{x}}\|) \tag{4}$$

- *a general subgradient of $f$ at $\bar{x}$ i.e., $\mathbf{u} \in \partial f(\mathbf{x})$, if there exist sequences $\mathbf{x}^\nu \xrightarrow{f} \bar{\mathbf{x}}$ (i.e., $\mathbf{x}^\nu \to \bar{\mathbf{x}}$ with $f(\mathbf{x}^\nu) \to f(\bar{\mathbf{x}})$) and $\mathbf{u}^\nu \in \hat{\partial} f(\mathbf{x}^\nu)$, with $\mathbf{u}^\nu \to \mathbf{u}$.*

- *a horizon subgradient of $f$ at $\bar{\mathbf{x}}$ i.e., $\mathbf{u} \in \partial^\infty f(\bar{\mathbf{x}})$ for some sequence $\lambda^\nu \searrow 0$, there exist a sequence of $\mathbf{u}^\nu \in \hat{\partial} f(\mathbf{x}^\nu)$ such that $\lambda^\nu \mathbf{u}^\nu \to \mathbf{u}$.*

*The sets $\hat{\partial} f(\bar{\mathbf{x}}), \partial f(\bar{\mathbf{x}}), \partial^\infty f(\bar{\mathbf{x}})$ are called regular, general and horizon subdifferential of $f$ at $\hat{\mathbf{x}}$, respectively.*

The above definitions implicitly assume that subgradients define hyperplanes that locally bound the function from below. Hence, they are also called as *lower* subgradients.

The following lemma (Theorem 7.1 of [1]) relates the set of general subgradients i.e., the general subdifferential, of an absolutely symmetric function $f$ with that of the corresponding singular value function.

**Lemma 2** (Theorem 7.1. of [1]). *Let $\mathbf{X} = \mathbf{U}\mathbf{\Sigma}\mathbf{V}^\top$ where $\mathbf{U} \in \mathbb{R}^{m \times r}, \mathbf{\Sigma} \in \mathbb{R}^{r \times r}$ and $\mathbf{V} \in \mathbb{R}^{n \times r}$, denote the singular value decomposition of $\mathbf{X} \in \mathbb{R}^{m \times n}$. The general subdifferential of a singular value function $f \circ \boldsymbol{\sigma}$ at $\mathbf{X}$ is given by the formula*

$$\partial(f \circ \boldsymbol{\sigma})(\mathbf{X}) = \mathbf{U}\operatorname{diag}(\partial f(\boldsymbol{\sigma}(\mathbf{X}))\mathbf{V}^\top \tag{5}$$

*where $\partial f(\boldsymbol{\sigma}(\mathbf{X}))$ is the general subdifferential of $f(\boldsymbol{\sigma}(\mathbf{X}))$. The regular and horizon subdifferential of $(f \circ \boldsymbol{\sigma})(\mathbf{X})$ i.e., $\hat{\partial}(f \circ \boldsymbol{\sigma})(\mathbf{X})$ and $\partial^\infty(f \circ \boldsymbol{\sigma})(\mathbf{X})$ can be similarly derived.*

Subderivatives generalize the notion of one-sided directional derivatives and are defined as follows

**Definition 4.** *Let $f : \mathbb{R}^q \to \bar{\mathbb{R}}$ where $\bar{\mathbb{R}} = \mathbb{R} \cup \{-\infty, +\infty\}$ and a point $\bar{\mathbf{x}}$ where $f(\bar{\mathbf{x}})$ is finite. The subderivative of $f$ at $\bar{\mathbf{x}}$ is defined as*

$$df_{\bar{\mathbf{x}}}(\bar{\mathbf{w}}) = \lim_{\substack{\tau \searrow 0 \\ \mathbf{w} \to \bar{\mathbf{w}}}} \inf \frac{f(\bar{\mathbf{x}} + \tau\mathbf{w}) - f(\bar{\mathbf{x}})}{\tau} \tag{6}$$

A critical property that ensures the dual relationship between regular subgradients and subderivatives is the so-called *subdifferential regularity*. The following lemma can be utilized for examining whether a function is subdifferentially regular or not.

**Lemma 3.** *( [2]) Let a function $f : \mathbb{R}^q \to \bar{\mathbb{R}}$ and a point $\bar{\mathbf{x}}$ with $f(\bar{\mathbf{x}})$ finite and $\partial f(\bar{\mathbf{x}}) \neq \emptyset$. $f$ is subdifferentially regular at $\bar{\mathbf{x}}$ if and only if $f$ is locally lower semi-continuous at $\bar{\mathbf{x}}$ with*

$$\partial f(\bar{\mathbf{x}}) = \hat{\partial} f(\bar{\mathbf{x}}) \quad \partial^\infty f(\bar{\mathbf{x}}) = \hat{\partial} f(\bar{\mathbf{x}})^\infty \tag{7}$$

*where $\hat{\partial} f(\bar{\mathbf{X}})^\infty$ denotes the horizon set of the set of the general subgradients of $f$.*

By Definition 3 we have that a vector $\mathbf{u} \in \mathbb{R}^q$ is a regular subgradient of $f(x)$ at $\bar{x}$ i.e., $u \in \hat{\partial} f(\bar{x})$ if

$$\lim_{x \to \bar{x}} \inf \frac{f(x) - f(\bar{x}) - \langle u, x - \bar{x} \rangle}{|x - \bar{x}|} \geq 0 \tag{8}$$

Clearly the regular subgradient of $f_i(x_i) = |x_i|^p$ for $x_i \in (-\infty, 0) \cup (0, +\infty)$ boils down to the gradients thereof hence the corresponding regular subdiferrential sets are singletons and coincide with the general ones. At $\bar{x} = 0$, the regular subdiferrential of $f_i$ is the interval $(-\infty, +\infty)$. That being said, we have

$$\hat{\partial} f_i = \begin{cases} -p\dfrac{1}{(-x_i)^{1-p}}, & x \in (-\infty, 0) \\ (-\infty, +\infty) & x_i = 0 \\ p\dfrac{1}{(x_i)^{1-p}}, & x \in (0, \infty) \end{cases} \tag{9}$$

The regular subdifferential of $\|\mathbf{x}\|_p^p$ can thus be obtained as $\partial\|\mathbf{x}\|_p^p = [\partial f_1(x_1), \partial f_2(x_2), \dots, \partial f_n(x_n)]$.

Lemma 3 is next used for proving subdifferential regularity of the $\ell_p$ quasi-norm with $p \in (0, 1)$.

**Lemma 4.** *The $\ell_p$ quasi-norm raised to the power of $p$ defined as $\|\mathbf{x}\|^p = \sum_{i=1}^q |x_i|^p$ where $\mathbf{x} \in \mathbb{R}^q$, is subdifferentially regular.*

*Proof.* Let us define the function $f_i(x_i) = |x_i|^p$. $\ell_p$ quasi-norm can be written as $\|\mathbf{x}\|_p^p = \sum_{i=1}^q f_i(x_i)$. Since the $\ell_p$ quasi-norm is separable, regular, general and horizon subgradient can be taken by cartesian products of the subgradients of $f_i$s, for $i = 1, 2, \ldots, q$. Moreover, $\|\mathbf{x}\|_p^p$ is regular at $\mathbf{x}$ when $f_i$s are regular at $x_i$s for all $i$s. Hence, we focus on $f_i(x_i) = |x_i|^p$. $f_i(x_i)$ is smooth and thus regular in $\mathbb{R} - \{0\}$. At 0, $f_i(0)$ is nondifferentiable and $\hat{\partial} f_i(0) = (-\infty, +\infty)$. Evidently, the set of regular subgradients is closed and coincides with $\partial f_i(0)$ (see Definition 3). Moreover, for the set of horizon subgradients at 0 we have $\partial^\infty f_i(0) = \{-\infty, +\infty\} \equiv \hat{\partial} f_i(0)^\infty$. Hence, due to Lemma 3 we can conclude the proof. $\qquad\square$

Having shown that the $\ell_p$ quasi-norm for $p \in (0, 1)$ is a subdifferentially regular, we can now show that the singular value function arising by the $\ell_p$ quasi-norm, i.e., the Schatten-$p$ quasi-norm is also subdifferentially regular.

**Lemma 5.** *The Schatten-p quasi-norm with $p \in (0, 1)$ is a subdifferentially regular function.*

*Proof.* See Corollary 7.5 of [1]. $\qquad\square$

Lemma 3 is critical in the proof of the Theorem 3 since it allows us to use the dual relationship between subderivatives and the regular subgradients of the Schatten-$p$ quasi-norm. The following theorem provides a dual correspondence between the general subgradients of a subdifferentially regular function and its subderivatives.

**Theorem 4** (Theorem 8.30 of [2]). *If a function $f$ is subdifferentially regular at $\bar{\mathbf{X}}$ then one has $\partial f(\bar{\mathbf{X}}) \neq \emptyset \leftrightarrow df(\bar{\mathbf{X}}) \neq -\infty$ and*

$$df_{\bar{\mathbf{X}}}(\mathbf{W}) = \sup\{\langle \mathbf{Q}, \mathbf{W}\rangle | \mathbf{Q} \in \partial f(\bar{\mathbf{X}})\} \tag{10}$$

*with $\partial f(\bar{\mathbf{X}})$ closed and convex.*

Next, the proof of Lemma 1, which gives the expressions for the regular subgradients of $\mathcal{S}_p$ quasi-norm raised to $p$ is provided.

## Proof of Lemma 1

*Proof.* From Lemma 2 we have that a matrix $\mathbf{Y}$ lies in $\hat{\partial}(f \circ \boldsymbol{\sigma})(\mathbf{X})$ if and only if $\boldsymbol{\sigma}(\mathbf{Y}) \in \hat{\partial} f(\boldsymbol{\sigma}(\mathbf{X}))$ and there exists a simultaneous singular value decomposition of the form $\mathbf{X} = \bar{\mathbf{U}} \operatorname{diag}(\boldsymbol{\sigma}(\mathbf{X})) \bar{\mathbf{V}}^\top$ and $\mathbf{Y} = \bar{\mathbf{U}} \operatorname{diag}(\boldsymbol{\sigma}(\mathbf{Y})) \bar{\mathbf{V}}^\top$. Note that we herein assume the full singular value decomposition i.e., matrices $\bar{\mathbf{U}}, \bar{\mathbf{V}}$ orthogonal of size $m \times m$ and $n \times n$, respectively. By representing $\bar{\mathbf{U}} = [\mathbf{U} \ \mathbf{U}_\perp]$ and $\bar{\mathbf{V}} = [\mathbf{V} \ \mathbf{V}_\perp]$ where $\mathbf{U}_\perp \in \mathbb{R}^{m \times m - r}$, $\mathbf{V}_\perp \in \mathbb{R}^{n \times n - r}$, $\boldsymbol{\Sigma}_+ = \operatorname{diag}(\boldsymbol{\sigma}_+(\mathbf{X}))$ and $\operatorname{colsp}(\mathbf{U}) \perp \operatorname{colsp}(\mathbf{U}_\perp)$, $\operatorname{colsp}(\mathbf{V}) \perp \operatorname{colsp}(\mathbf{V}_\perp)$, we rewrite the singular value decomposition of $\mathbf{X}$ in the form

$$\mathbf{X} = [\mathbf{U} \ \mathbf{U}_\perp] \begin{bmatrix} \boldsymbol{\Sigma}_+ & \mathbf{0} \\ \mathbf{0} & \mathbf{0} \end{bmatrix} \begin{bmatrix} \mathbf{V} \\ \mathbf{V}_\perp \end{bmatrix} \tag{11}$$

Note that since $\mathbf{U}_\perp$ and $\mathbf{V}_\perp$ correspond to zero singular values of $\mathbf{X}$, they are not uniquely defined. Next, going back to the form of a regular subgradient $\mathbf{Y}$ of $\|\mathbf{X}\|_{\mathcal{S}_p}^p$ and based on the above-defined expression of the full singular value decomposition of $\mathbf{X}$ we have,

$$\mathbf{Y} = [\mathbf{U} \ \mathbf{U}_\perp] \begin{bmatrix} \hat{\partial} \|\boldsymbol{\Sigma}_+(\mathbf{X})\|_p^p & \mathbf{0} \\ \mathbf{0} & \mathbf{D} \end{bmatrix} \begin{bmatrix} \mathbf{V} \\ \mathbf{V}_\perp \end{bmatrix} \tag{12}$$

where $\mathbf{D}$ contain elements in $(-\infty, +\infty)$ on its main diagonal and zeros elsewhere. eq. (12) can be written in a more compact form as $\mathbf{Y} = \mathbf{U} \hat{\partial} \|\boldsymbol{\Sigma}_+(\mathbf{X})\|_p^p \mathbf{V}^\top + \mathbf{U}_\perp \mathbf{D} \mathbf{V}_\perp^\top$ and by setting $\mathbf{W} = \mathbf{U}_\perp \mathbf{D} \mathbf{V}_\perp^\top$, we get the expression for the regular subdifferential of $\|\mathbf{X}\|_{\mathcal{S}_p}^p$. $\qquad\square$

**Lemma 6.** *Let $(\hat{\mathbf{U}}, \hat{\mathbf{V}})$ with $\hat{\mathbf{U}} = \mathbf{U}\boldsymbol{\Sigma}^{\frac{1}{2}}$ and $\hat{\mathbf{V}} = \mathbf{V}\boldsymbol{\Sigma}^{\frac{1}{2}}$ be a stationary point of (3), where non all-zero columns of matrices $\mathbf{U} \in \mathbb{R}^{m \times d}$ and $\mathbf{V} \in \mathbb{R}^{n \times d}$ are orthonormal, and $\boldsymbol{\Sigma}$ is a $d \times d$ real-valued*

*diagonal matrix. The pair* $(\hat{\mathbf{U}}, \hat{\mathbf{V}})$ *satisfies the following equations*

$$\nabla l(\mathbf{Y}, \hat{\mathbf{U}}\hat{\mathbf{V}}^\top)\hat{\mathbf{V}} + \hat{\mathbf{U}}\partial\|\mathbf{\Sigma}\|_{\mathcal{S}_p}^p \ni 0 \tag{13}$$

$$\nabla l(\mathbf{Y}, \hat{\mathbf{U}}^\top\hat{\mathbf{V}}^\top)^\top\hat{\mathbf{U}} + \hat{\mathbf{V}}\partial\|\mathbf{\Sigma}\|_{\mathcal{S}_p}^p \ni 0 \tag{14}$$

*Proof.* Since $(\hat{\mathbf{U}}, \hat{\mathbf{V}})$ is a local minimum of the objective defined in (3), it is also a stationary point of it.

Moreover, by defining $f_i(x) = x_i^p$ we can get

$$\partial_{\mathbf{u}_i} \sum_{i=1}^d f_i\left(\frac{1}{2}(\|\mathbf{u}_i\|_2^2 + \|\mathbf{v}_i\|_2^2)\right) = \mathbf{u}_i \partial f_i\left(\frac{1}{2}(\|\mathbf{u}_i\|_2^2 + \|\mathbf{v}_i\|_2^2)\right) \tag{15}$$

Hence, by using the chain product rule w.r.t. to $\mathbf{U}$ we can get

$$\partial_{\mathbf{U}} \sum_{i=1}^d f_i\left(\frac{1}{2}(\|\mathbf{u}_i\|_2^2 + \|\mathbf{v}_i\|_2^2)\right) = \mathbf{U}\,\mathrm{diag}\left[\partial f_i\left(\frac{1}{2}(\|\mathbf{u}_i\|_2^2 + \|\mathbf{v}_i\|_2^2)\right)\right]_{i=1,2\ldots,d} \tag{16}$$

(16) estimated at $(\hat{\mathbf{U}}, \hat{\mathbf{V}})$ becomes

$$\partial_{\mathbf{U}} \sum_{i=1}^d f_i\left(\frac{1}{2}(\|\mathbf{u}_i\|_2^2 + \|\mathbf{v}_i\|_2^2)\right)\bigg|_{(\hat{\mathbf{U}},\hat{\mathbf{V}})} = \hat{\mathbf{U}}\,\mathrm{diag}[\partial f_i(\sigma_i)]_{i=1,2,\ldots,d} \equiv \hat{\mathbf{U}}\partial\|\mathbf{\Sigma}\|_{\mathcal{S}_p}^p \tag{17}$$

In addition to eq. (16), we have

$$\nabla_{\mathbf{U}} l(\mathbf{Y}, \mathbf{U}\mathbf{V}^\top) = \nabla l(\mathbf{Y}, \mathbf{U}\mathbf{V}^\top)\mathbf{V} \tag{18}$$

Eq. (17) and (18) give rise to the following condition for the stationary point $(\hat{\mathbf{U}}, \hat{\mathbf{V}})$ of (3)

$$\nabla l(\mathbf{Y}, \hat{\mathbf{U}}\hat{\mathbf{V}}^\top)\hat{\mathbf{V}} + \hat{\mathbf{U}}\partial\|\mathbf{\Sigma}\|_{\mathcal{S}_p}^p \ni 0 \tag{19}$$

Following a similar process for partial gradients w.r.t $\mathbf{V}$ we get the second stationary point condition for $\mathcal{L}_2$

$$\nabla l(\mathbf{Y}, \hat{\mathbf{U}}^\top\hat{\mathbf{V}}^\top)^\top\hat{\mathbf{U}} + \hat{\mathbf{V}}\partial\|\mathbf{\Sigma}\|_{\mathcal{S}_p}^p \ni 0 \tag{20}$$

□

Next we move on the proof of Theorem 3. The proof builds upon and significantly extends the results presented in Theorem 2 of [3]. In particular, contrary to the latter which applies to a specific regularizer characterized by some convenient properties (e.g. weak convexity), in our case by using the dual correspondence between subderivatives and regular subgradients, we can get similar results for any concave singular value penalty function as long as it is *subdifferentially regular*. Note that subdifferential regularity is a quite general condition and ensures that the set regular subgradients coincides with the set of the general ones. Moreover, Theorem 3 can be considered as an enhanced version of Theorem 2 of [3], since it that local minima of the variational $\mathcal{S}_p$ regularized problem correspond to local minima of the original problem defined over $\mathbf{X}$. That said, it goes one step further form Theorem 2 of [3] which shows that directional derivatives are nonnegative w.r.t. low-rank perturbations.

## Proof of Theorem 3

*Proof.* Since $(\hat{\mathbf{U}}, \hat{\mathbf{V}})$ is a local minimizer of $\mathcal{L}_1(\mathbf{U}, \mathbf{V})$ and $\mathcal{L}_2(\mathbf{U}, \mathbf{V})$ defined in (2) and (3), respectively, we can define a small perturbation of $(\hat{\mathbf{U}}, \hat{\mathbf{V}})$ i.e, $(\mathbf{U}_t, \mathbf{V}_t)$

$$\mathbf{U}_t\mathbf{V}_t^\top = \hat{\mathbf{U}}\hat{\mathbf{V}}^\top + t\tilde{\mathbf{U}}\tilde{\mathbf{V}}^\top \tag{21}$$

with $\tilde{\mathbf{U}} \in \mathbb{R}^{m \times m}, \tilde{\mathbf{V}} \in \mathbb{R}^{n \times m}$ assuming that $m \leq n$, such that

$$\mathcal{L}_i(\mathbf{U}_t, \mathbf{V}_t) - \mathcal{L}_i(\hat{\mathbf{U}}, \hat{\mathbf{V}}) \geq 0, \tag{22}$$

for $i = 1, 2$ and $t \searrow 0$.

Moreover, since $\hat{\mathbf{U}}$ and $\hat{\mathbf{V}}$ are defined as $\hat{\mathbf{U}} = \mathbf{U}\boldsymbol{\Sigma}^{\frac{1}{2}}$ and $\mathbf{V}\boldsymbol{\Sigma}^{\frac{1}{2}}$ where $\mathbf{U}, \mathbf{V}$ are of size $m \times m$ and $n \times m$, respectively, contain orthonormal columns, and $\boldsymbol{\Sigma}$ is a diagonal $m \times m$ matrix, we have

$$\mathcal{L}_1(\hat{\mathbf{U}}, \hat{\mathbf{V}}) = \mathcal{L}_2(\hat{\mathbf{U}}, \hat{\mathbf{V}}) = \mathcal{L}(\hat{\mathbf{X}}) \tag{23}$$

Let us define

$$\hat{\mathbf{X}} + t\tilde{\mathbf{X}} = \mathbf{U}_t \mathbf{V}_t^\top \tag{24}$$

where the product $\tilde{\mathbf{U}}\tilde{\mathbf{V}}^\top$ of (21) has been replaced by $\tilde{\mathbf{X}}$.

Our goal is to show that $\hat{\mathbf{X}}$ is local minimum of $\mathcal{L}(\mathbf{X})$ i.e., $\mathcal{L}(\hat{\mathbf{X}} + t\tilde{\mathbf{X}}) - \mathcal{L}(\hat{\mathbf{X}}) \geq 0$ for $t \searrow 0$. Due to (22) and (23) the result immediately arises if we show that $\mathcal{L}(\hat{\mathbf{X}} + t\tilde{\mathbf{X}}) = \mathcal{L}_i(\mathbf{U}_t, \mathbf{V}_t)$ for $i = 1, 2$. $\mathcal{L}(\mathbf{X})$ consists of the sum of a differentiable loss function denoted as $l(\mathbf{Y}, \mathbf{X})$ and the nonconvex matrix function $\|\mathbf{X}\|_{\mathcal{S}_p}^p$ and its subderivatives at $\hat{\mathbf{X}}$ w.r.t $\tilde{\mathbf{X}}$ are defined as

$$d\mathcal{L}_{\tilde{\mathbf{X}}}(\hat{\mathbf{X}}) = \lim_{\substack{t \searrow 0 \\ \mathbf{Z} \to \tilde{\mathbf{X}}}} \inf \frac{\mathcal{L}(\hat{\mathbf{X}} + t\mathbf{Z}) - \mathcal{L}(\hat{\mathbf{X}})}{t} \tag{25}$$

From the above definition and due to the continuity of $\mathcal{L}(\mathbf{X})$ it becomes evident that nonnegative subderivatives imply $\mathcal{L}(\hat{\mathbf{X}} + t\tilde{\mathbf{X}}) - \mathcal{L}(\hat{\mathbf{X}}) \geq 0$ for $t \searrow 0$.

Next, without loss of generality we assume, that the columns of $\tilde{\mathbf{U}}$ (and $\tilde{\mathbf{V}}$) belong to the subspace formed by the direct sum of the columnspace of $\hat{\mathbf{U}}$ (resp. $\hat{\mathbf{V}}$) and the columnspace of a matrix $\hat{\mathbf{U}}_\perp \in \mathbb{R}^{m \times (m-r)}$ (resp. $\hat{\mathbf{V}}_\perp \in \mathbb{R}^{n \times (n-r)}$) with $\mathrm{rank}(\hat{\mathbf{U}}_\perp) \leq (m-r)$ (resp. $\mathrm{rank}(\hat{\mathbf{V}}_\perp) \leq (n-r)$) such that $\mathrm{colsp}(\hat{\mathbf{U}}) \perp \mathrm{colsp}(\hat{\mathbf{U}}_\perp)$ (resp. $\mathrm{colsp}(\hat{\mathbf{V}}) \perp \mathrm{colsp}(\hat{\mathbf{V}}_\perp)$ ). That said we get

$$\tilde{\mathbf{U}} = \hat{\mathbf{U}}\mathbf{A} + \hat{\mathbf{U}}_\perp \mathbf{B} \tag{26}$$

$$\tilde{\mathbf{V}} = \hat{\mathbf{V}}\mathbf{C} + \hat{\mathbf{V}}_\perp \mathbf{D} \tag{27}$$

Since matrices $\mathbf{A}, \mathbf{C}$ of size $m \times m$ and $\mathbf{B}, \mathbf{D}$ of sizes $(m - r) \times m$ and $(n - r) \times m$ respectively, are arbitrary, there is no loss of generality by expressing $\tilde{\mathbf{U}}, \tilde{\mathbf{V}}$ as in (26) and (27). By (26) and (27) we hence have

$$\tilde{\mathbf{U}}\tilde{\mathbf{V}}^\top = \hat{\mathbf{U}}\underbrace{\mathbf{A}\mathbf{C}^\top}_{\mathbf{K}_1}\hat{\mathbf{V}}^\top + \hat{\mathbf{U}}\underbrace{\mathbf{A}\mathbf{D}^\top}_{\mathbf{K}_2}\hat{\mathbf{V}}_\perp^\top + \hat{\mathbf{U}}_\perp\underbrace{\mathbf{B}\mathbf{C}^\top}_{\mathbf{K}_3}\hat{\mathbf{V}}^\top + \hat{\mathbf{U}}_\perp\underbrace{\mathbf{B}\mathbf{D}^\top}_{\mathbf{K}_4}\hat{\mathbf{V}}_\perp^\top \tag{28}$$

$$= \hat{\mathbf{U}}\mathbf{K}_1\hat{\mathbf{V}}^\top + \hat{\mathbf{U}}\mathbf{K}_2\hat{\mathbf{V}}_\perp^\top + \hat{\mathbf{U}}_\perp\mathbf{K}_3\hat{\mathbf{V}}^\top + \hat{\mathbf{U}}_\perp\mathbf{K}_4\hat{\mathbf{V}}_\perp^\top \tag{29}$$

By using Theorem 4, we can employ the dual relationship between the subderivatives of $\mathcal{L}(\mathbf{X})$ defined in (25) and the regular subgradients thereof,

$$d\mathcal{L}_{\tilde{\mathbf{X}}}(\hat{\mathbf{X}}) = \sup\{\langle \mathbf{Q}, \tilde{\mathbf{X}} \rangle | \mathbf{Q} \in \partial\|\hat{\mathbf{X}}\|_{\mathcal{S}_p}^p\} + \langle \nabla l(\mathbf{Y}, \hat{\mathbf{X}}), \tilde{\mathbf{X}} \rangle \tag{30}$$

In the following, we will show that the values of the subderivatives in (30), remain unaffected if we assume that $\mathbf{K}_1$ is a diagonal matrix and matrices $\mathbf{K}_2, \mathbf{K}_3$ vanish. In doing so, both matrices $\tilde{\mathbf{U}}$ and $\tilde{\mathbf{V}}$ are conveniently expressed in a desirable and simplified form.

Focusing on the first term of (30) and by using the form for the subgradients of $\|\mathbf{X}\|_{\mathcal{S}_p}^p$ given in Lemma 1 at $\hat{\mathbf{X}}$ where $\hat{\mathbf{X}} = \mathbf{U}_{\mathbf{X}}\boldsymbol{\Sigma}_+\mathbf{V}_{\mathbf{X}}$, with $\mathbf{U}_{\mathbf{X}} \in \mathbb{R}^{m \times r}, \mathbf{V}_{\mathbf{X}} \in \mathbb{R}^{n \times r}$ and $\boldsymbol{\Sigma}_+ \in \mathbb{R}_+^{r \times r}$ denotes the singular value decomposition of $\hat{\mathbf{X}}$ we have,

$$\langle \mathbf{U}_{\mathbf{X}}\partial\|\boldsymbol{\Sigma}_+\|_p^p\mathbf{V}_{\mathbf{X}}^\top + \mathbf{W}, \tilde{\mathbf{U}}\tilde{\mathbf{V}}^\top \rangle = \langle \mathbf{U}_{\mathbf{X}}\partial\|\boldsymbol{\Sigma}_+\|_p^p\mathbf{V}_{\mathbf{X}}^\top, \tilde{\mathbf{U}}\tilde{\mathbf{V}}^\top \rangle + \langle \mathbf{W}, \tilde{\mathbf{U}}\tilde{\mathbf{V}}^\top \rangle \tag{31}$$

By using (29), the first term of (31) is rewritten as

$$\langle \mathbf{U}_{\mathbf{X}}\partial\|\boldsymbol{\Sigma}_+\|_p^p\mathbf{V}_{\mathbf{X}}^\top, \tilde{\mathbf{U}}\tilde{\mathbf{V}}^\top \rangle = \langle \mathbf{U}_{\mathbf{X}}\partial\|\boldsymbol{\Sigma}_+\|_p^p\mathbf{V}_{\mathbf{X}}^\top, \hat{\mathbf{U}}\mathbf{K}_1\hat{\mathbf{V}}^\top + \hat{\mathbf{U}}\mathbf{K}_2\hat{\mathbf{V}}_\perp^\top + \hat{\mathbf{U}}_\perp\mathbf{K}_3\hat{\mathbf{V}}^\top + \hat{\mathbf{U}}_\perp\mathbf{K}_4\hat{\mathbf{V}}_\perp^\top \rangle$$
$$= \langle \mathbf{U}_{\mathbf{X}}\partial\|\boldsymbol{\Sigma}_+\|_p^p\mathbf{V}_{\mathbf{X}}^\top, \hat{\mathbf{U}}\mathbf{K}_1\hat{\mathbf{V}}^\top \rangle = \langle \partial\|\boldsymbol{\Sigma}_+\|_p^p, \mathbf{U}_{\mathbf{X}}^\top\hat{\mathbf{U}}\mathbf{K}_1\hat{\mathbf{V}}^\top\mathbf{V}_{\mathbf{X}} \rangle \tag{32}$$

We have $\mathbf{U}_{\mathbf{X}}^\top \hat{\mathbf{U}} = [\boldsymbol{\Sigma}_+^{\frac{1}{2}}\ \mathbf{0}_{m-r}]$ and $\mathbf{V}_{\mathbf{X}}^\top \hat{\mathbf{V}} = [\boldsymbol{\Sigma}_+^{\frac{1}{2}}\ \mathbf{0}_{m-r}]$ where $\mathbf{0}_{m-r}$ denote $m-r$ zero columns, and next we define with a slight abuse of notation $\tilde{\boldsymbol{\Sigma}}^{\frac{1}{2}} = [\boldsymbol{\Sigma}_+^{\frac{1}{2}}\ \mathbf{0}_{m-r}]$. From (32) we have

$$\langle \mathbf{U}_{\mathbf{X}} \partial \|\boldsymbol{\Sigma}_+\|_p^p \mathbf{V}_{\mathbf{X}}^\top, \tilde{\mathbf{U}}\tilde{\mathbf{V}}^\top \rangle = \langle \partial \|\boldsymbol{\Sigma}_+\|_p^p, \tilde{\boldsymbol{\Sigma}}^{\frac{1}{2}} \mathbf{K}_1 \tilde{\boldsymbol{\Sigma}}^{T,\frac{1}{2}} \rangle \tag{33}$$

Based on the above analysis, it becomes evident that (33) is invariant to the values of non-diagonal elements of $\mathbf{K}_1$. Finally, for the second term of (31), we have

$$\langle \mathbf{W}, \tilde{\mathbf{U}}\tilde{\mathbf{V}}^\top \rangle = \langle \mathbf{W}, \hat{\mathbf{U}}\mathbf{K}_1\hat{\mathbf{V}}^\top + \hat{\mathbf{U}}\mathbf{K}_2\hat{\mathbf{V}}_\perp^\top + \hat{\mathbf{U}}_\perp\mathbf{K}_3\hat{\mathbf{V}}^\top + \hat{\mathbf{U}}_\perp\mathbf{K}_4\hat{\mathbf{V}}_\perp^\top \rangle = \langle \mathbf{W}, \hat{\mathbf{U}}_\perp\mathbf{K}_4\hat{\mathbf{V}}_\perp^\top \rangle$$
$$= \langle \mathbf{D}, \mathbf{K}_4 \rangle \tag{34}$$

where the last equality by definition of $\mathbf{W}$ according to Lemma 1. Recall that $\mathbf{D} \in \mathbb{R}^{(m-r)\times(n-r)}$ contains elements in the interval $(-\infty, +\infty)$ on its main diagonal and zeros elsewhere and hence the term $\mathbf{D}, \mathbf{K}_4$, is also invariant to non-diagonal elements of $\mathbf{K}_4$.

Let us now focus on the second term of the subderivative defined in (30) i.e., $\langle \nabla l(\mathbf{Y}, \hat{\mathbf{X}}), \tilde{\mathbf{X}} \rangle$. We have

$$\langle \nabla l(\mathbf{Y}, \hat{\mathbf{U}}\hat{\mathbf{V}}^\top), \tilde{\mathbf{U}}\tilde{\mathbf{V}}^\top \rangle = \langle \nabla l(\mathbf{Y}, \hat{\mathbf{U}}\hat{\mathbf{V}}^\top), \hat{\mathbf{U}}\mathbf{K}_1\hat{\mathbf{V}}^\top + \hat{\mathbf{U}}\mathbf{K}_2\hat{\mathbf{V}}_\perp^\top + \hat{\mathbf{U}}_\perp\mathbf{K}_3\hat{\mathbf{V}}^\top + \hat{\mathbf{U}}_\perp\mathbf{K}_4\hat{\mathbf{V}}_\perp^\top \rangle \tag{35}$$

$$= \langle \nabla l(\mathbf{Y}, \hat{\mathbf{U}}\hat{\mathbf{V}}^\top), \hat{\mathbf{U}}\mathbf{K}_1\hat{\mathbf{V}}^\top \rangle + \langle \nabla l(\mathbf{Y}, \hat{\mathbf{U}}\hat{\mathbf{V}}^\top), \hat{\mathbf{U}}\mathbf{K}_2\hat{\mathbf{V}}_\perp^\top \rangle$$
$$+ \langle \nabla l(\mathbf{Y}, \hat{\mathbf{U}}\hat{\mathbf{V}}^\top), \hat{\mathbf{U}}_\perp\mathbf{K}_3\hat{\mathbf{V}}^\top \rangle + \langle \nabla l(\mathbf{Y}, \hat{\mathbf{U}}\hat{\mathbf{V}}^\top), \hat{\mathbf{U}}_\perp\mathbf{K}_4\hat{\mathbf{V}}_\perp^\top \rangle \tag{36}$$

Focusing on the first term of (36) we get

$$\langle \nabla l(\mathbf{Y}, \hat{\mathbf{U}}\hat{\mathbf{V}}^\top), \hat{\mathbf{U}}\mathbf{K}_1\hat{\mathbf{V}}^\top \rangle = \langle \nabla l(\mathbf{Y}, \hat{\mathbf{V}}^\top)\hat{\mathbf{V}}, \hat{\mathbf{U}}\mathbf{K}_1\hat{\mathbf{V}} \rangle \tag{37}$$

(37) due to Lemma 6 takes the form

$$\langle \nabla l(\mathbf{Y}, \hat{\mathbf{V}}^\top)\hat{\mathbf{V}}, \hat{\mathbf{U}}\mathbf{K}_1\hat{\mathbf{V}} \rangle = -\langle \hat{\mathbf{U}}\partial \|\boldsymbol{\Sigma}\|_{\mathcal{S}_p}^p, \hat{\mathbf{U}}\mathbf{K}_1 \rangle = -\langle \partial \|\boldsymbol{\Sigma}\|_{\mathcal{S}_p}^p, \boldsymbol{\Sigma}\mathbf{K}_1 \rangle \tag{38}$$

Following a similar analysis as above we can easily see that (38) is likewise invariant to non-diagonal elements of $\mathbf{K}_1$. For the second term of (37) and again due to Lemma 6 we have

$$\langle \nabla l(\mathbf{Y}, \hat{\mathbf{U}}\hat{\mathbf{V}}^\top), \hat{\mathbf{U}}\mathbf{K}_2\hat{\mathbf{V}}_\perp^\top \rangle = -\langle \hat{\mathbf{V}}\partial \|\boldsymbol{\Sigma}\|_{\mathcal{S}_p}^p, \mathbf{K}_2\hat{\mathbf{V}}_\perp^\top \rangle = 0 \tag{39}$$

It can be easily noticed that the same holds for the third term of (36) involving $\mathbf{K}_3$. Hence we have shown that the values of the subderivatives defined in (25) for the given $\tilde{\mathbf{X}} = \tilde{\mathbf{U}}\tilde{\mathbf{V}}^\top$ where $\tilde{\mathbf{U}}\tilde{\mathbf{V}}^\top$ takes the form given in (29), are not affected by $\mathbf{K}_2, \mathbf{K}_3$ and non-diagonal elements of $\mathbf{K}_1$.

That being said we can now simplify the expression of $\tilde{\mathbf{X}} = \tilde{\mathbf{U}}\tilde{\mathbf{V}}^\top$ without loosing generality as

$$\tilde{\mathbf{U}}\tilde{\mathbf{V}}^\top = \mathbf{U}_{\mathbf{X}}\mathbf{T}\mathbf{V}_{\mathbf{X}}^\top + \mathbf{U}_{\mathbf{X},\perp}\mathbf{P}\mathbf{V}_{\mathbf{X},\perp}^T \tag{40}$$

where $\mathbf{U}_{\mathbf{X},\perp}\mathbf{P}\mathbf{V}_{\mathbf{X},\perp}^T$ arises by the singular value decomposition of $\hat{\mathbf{U}}_\perp\mathbf{K}_4\hat{\mathbf{V}}_\perp^\top$. Note that $\mathbf{T}, \mathbf{P}$ are $r \times r$ and $m-r \times m-r$ diagonal matrices containing the $r$ nonzero diagonal elements of $\mathbf{K}_1\boldsymbol{\Sigma}$ and the singular values of $\hat{\mathbf{U}}_\perp^\top\mathbf{K}_4\hat{\mathbf{V}}_\perp^\top$, respectively. With the simplified form of (40) we now go back to the expression of $\mathbf{U}_t\mathbf{V}_t^\top$ in (24) assuming $t \to 0$,

$$\mathbf{U}_t\mathbf{V}_t^\top = \hat{\mathbf{U}}\hat{\mathbf{V}}^\top + t\tilde{\mathbf{U}}\tilde{\mathbf{V}}^\top \tag{41}$$

$$= \mathbf{U}_{\mathbf{X}}\boldsymbol{\Sigma}_+\mathbf{V}_{\mathbf{X}}^\top + t(\mathbf{U}_{\mathbf{X}}\mathbf{T}\mathbf{V}_{\mathbf{X}}^\top + \mathbf{U}_{\mathbf{X},\perp}\mathbf{P}\mathbf{V}_{\mathbf{X},\perp}^\top) \tag{42}$$

$$= [\mathbf{U}_{\mathbf{X}}\ \mathbf{U}_{\mathbf{X},\perp}] \begin{bmatrix} \boldsymbol{\Sigma}_+ + t\mathbf{T} & 0 \\ 0 & t\mathbf{P} \end{bmatrix} \begin{bmatrix} \mathbf{V}_{\mathbf{X}} \\ \mathbf{V}_{\mathbf{X},\perp} \end{bmatrix} \tag{43}$$

(43) can be viewed as the singular value decomposition of matrix $\hat{\mathbf{X}} + t\tilde{\mathbf{X}}$.

By setting $\mathbf{U}_t = [\mathbf{U}_{\mathbf{X}}\ \mathbf{U}_{\mathbf{X},\perp}] \begin{bmatrix} (\boldsymbol{\Sigma}_+ + t\mathbf{T})^{\frac{1}{2}} & 0 \\ 0 & (t\mathbf{P})^{\frac{1}{2}} \end{bmatrix}$

and $\mathbf{V}_t = [\mathbf{V}_{\mathbf{X}}\ \mathbf{V}_{\mathbf{X},\perp}] \begin{bmatrix} (\boldsymbol{\Sigma}_+ + t\mathbf{T})^{\frac{1}{2}} & 0 \\ 0 & (t\mathbf{P})^{\frac{1}{2}} \end{bmatrix}$ it becomes evident that $\mathcal{L}(\hat{\mathbf{X}} + t\tilde{\mathbf{X}}) = \mathcal{L}_i(\mathbf{U}_t, \mathbf{V}_t)$
for $i = 1, 2$, which concludes the proof. $\square$

## 2 Proof of Proposition 1

*Proof.* If we first consider the case with $\delta = 0$, then since $\|\mathbf{u}\|_2 = 1$ and $\|\mathbf{v}\|_2 = 1$ we need to solve the following

$$\arg\min_{\tau \geq 0, \mathbf{u}, \mathbf{v}} \left\{ f(\tau, \mathbf{u}, \mathbf{v}) = -\tau^2 \langle \mathcal{A}^*(\mathbf{R}), \mathbf{uv}^\top \rangle + \tfrac{1}{2}\tau^4 + \lambda\tau^{2p} \right\} \quad \text{s.t.} \quad \|\mathbf{u}\|_2 = 1, \; \|\mathbf{v}\|_2 = 1 \quad (44)$$

Note that for any non-negative $\tau$ the solution to the above w.r.t. $(\mathbf{u}, \mathbf{v})$ is given by the largest singular vector pair of $\mathcal{A}^*(\mathbf{R})$. Substituting this into the above equation gives the following problem:

$$\arg\min_{\tau \geq 0} -\tau^2 \sigma(\mathcal{A}^*(\mathbf{R})) + \tfrac{1}{2}\tau^4 + \lambda\tau^{2p} \quad (45)$$

Clearly for $\tau = 0$ the above equation is equal to 0, so we are left to test whether $\exists \tau > 0$ such that the above is strictly less than 0. For $p \in (0, 1)$, $\tau > 0$, this results in the following:

$$-\tau^2\sigma(\mathcal{A}^*(\mathbf{R})) + \tfrac{1}{2}\tau^4 + \lambda\tau^{2p} < 0 \iff \lambda - \tau^{2-2p}\sigma(\mathcal{A}^*(\mathbf{R})) + \tfrac{1}{2}\tau^{4-2p} < 0 \quad (46)$$

Finding the critical points of the above w.r.t. $\tau$ we get:

$$-(2 - 2p)\tau^{1-2p}\sigma(\mathcal{A}^*(\mathbf{R})) + \tfrac{4-2p}{2}\tau^{3-2p} = 0 \iff \quad (47)$$

$$-(2 - 2p)\sigma(\mathcal{A}^*(\mathbf{R})) + (2 - p)\tau^2 = 0 \iff \quad (48)$$

$$\tau = \pm\sqrt{\frac{2-2p}{2-p}\sigma(A^*(\mathbf{R}))} \quad (49)$$

Since only the positive root is feasible we can substitute it into (46) to test whether the minimum is strictly negative.

To see the result with $\delta > 0$, note that result is given follow the same approach, with the exception that the objective in (44) is no longer the exact objective we need minimize, but rather we need to minimize:

$$\tilde{f}(\tau, \mathbf{u}, \mathbf{v}) = -\tau^2\langle \mathcal{A}^*(\mathbf{R}), \mathbf{uv}^\top \rangle + \tfrac{1}{2}\tau^4\|\mathcal{A}(\mathbf{uv}^\top)\|_F^2 + \lambda\tau^{2p} \quad (50)$$

The result is then completed by noting that for $\|\mathbf{u}\|_2 = \|\mathbf{v}\|_2 = 1$ and any $\tau \geq 0$ the following bound is provided by restricted isometry:

$$|\tilde{f}(\tau, \mathbf{u}, \mathbf{v}) - f(\tau, \mathbf{u}, \mathbf{v})| = |\tfrac{1}{2}\tau^4\|\mathcal{A}(\mathbf{uv}^\top)\|_F^2 - \tfrac{1}{2}\tau^4| \leq \tfrac{1}{2}\delta\tau^4 \quad (51)$$

$\square$

## 3 The proposed Variational Schatten-$p$ matrix completion algorithm

In this section we present the matrix completion minimization algorithm for variational $\mathcal{S}_p$ regularized objective function.

Recall that the factorized $\mathcal{S}_p$ regularized objective function is given as,

$$\min_{\mathbf{U}, \mathbf{V}} \; \mathcal{L}(\mathbf{U}, \mathbf{V}), \quad \text{where} \quad (52)$$

$$\mathcal{L}(\mathbf{U}, \mathbf{V}) = \frac{1}{2}\|\mathcal{P}_Z(\mathbf{Y} - \mathbf{UV}^T)\|_F^2 + \lambda\sum_{i=1}^{d}(\|\mathbf{u}_i\|_2^2 + \|\mathbf{v}_i\|_2^2)^p. \quad (53)$$

The proposed Variational Schatten-$p$ algorithm is based on the ideas stemming from the block successive minimization framework (BSUM), [4, 5]. That is, it assumes local tight upper-bounds of the objective (53), for updating the matrix factors $\mathbf{U}$ and $\mathbf{V}$. More specifically, following the relevant low-rank matrix factorization based matrix completion algorithm of [6], matrices $\mathbf{U}$ and $\mathbf{V}$ are updated by minimizing approximate second order Taylor expansions of (53), i.e.,

$$\mathbf{U}^{t+1} = \min_{\mathbf{U}} g(\mathbf{U}|\mathbf{U}^t, \mathbf{V}^t) \quad (54)$$

and

$$\mathbf{V}^{t+1} = \min_{\mathbf{V}} q(\mathbf{V}|\mathbf{U}^{t+1}, \mathbf{V}^t) \quad (55)$$

where the superscript $t$ denotes the iteration number and $g(\mathbf{U}|\mathbf{U}^t, \mathbf{V}^t)$ and $q(\mathbf{V}|\mathbf{U}^{t+1}, \mathbf{V}^t)$ have the following form

$$
g(\mathbf{U}|\mathbf{U}^t, \mathbf{V}^t) = \mathcal{L}(\mathbf{U}^t, \mathbf{V}^t) + \text{trace}\{(\mathbf{U} - \mathbf{U}^t)^\top \nabla_\mathbf{U} \mathcal{L}(\mathbf{U}^t, \mathbf{V}^t)\}
$$

$$
+ \frac{1}{2}\text{vec}(\mathbf{U} - \mathbf{U}^t)^\top \bar{\mathbf{H}}_{\mathbf{U}^t} \text{vec}(\mathbf{U} - \mathbf{U}^t) \tag{56}
$$

$$
q(\mathbf{V}|\mathbf{U}^{t+1}, \mathbf{V}^t) = \mathcal{L}(\mathbf{U}^{t+1}, \mathbf{V}^t) + \text{trace}\{(\mathbf{V} - \mathbf{V}^t)^\top \nabla_\mathbf{V} \mathcal{L}(\mathbf{U}^{t+1}, \mathbf{V}^t)\}
$$

$$
+ \frac{1}{2}\text{vec}(\mathbf{V} - \mathbf{V}^t)^\top \bar{\mathbf{H}}_{\mathbf{V}^t} \text{vec}(\mathbf{V} - \mathbf{V}^t) \tag{57}
$$

The approximate Hessian matrices $\bar{\mathbf{H}}_{\mathbf{U}^t}$ and $\bar{\mathbf{H}}_{\mathbf{V}^t}$ are of block diagonal form of sizes $md \times md$ and $nd \times nd$, respectively, and are defined as follows

$$
\bar{\mathbf{H}}_{\mathbf{U}^t} = \mathbf{I}_m \otimes \tilde{\mathbf{H}}_{\mathbf{U}^t} \tag{58}
$$

$$
\bar{\mathbf{H}}_{\mathbf{V}^t} = \mathbf{I}_n \otimes \tilde{\mathbf{H}}_{\mathbf{V}^t} \tag{59}
$$

where $\otimes$ denotes the Kronecker product,

$$
\tilde{\mathbf{H}}_{\mathbf{U}^t} = \mathbf{V}^{t,\top} \mathbf{V}^t + \lambda \mathbf{W}_{(\mathbf{U}^t, \mathbf{V}^t)} \tag{60}
$$

$$
\tilde{\mathbf{H}}_{\mathbf{V}^t} = \mathbf{U}^{t+1,\top} \mathbf{U}^{t+1} + \lambda \mathbf{W}_{(\mathbf{U}^{t+1}, \mathbf{V}^t)} \tag{61}
$$

and $\mathbf{W}_{(\mathbf{U}^t, \mathbf{V}^t)} = \text{diag}([p(\|\mathbf{u}_i\|_2^2 + \|\mathbf{v}_i\|_2^2)^{p-1}]_{i=1,2,\ldots,d})$. Since the objective function is non-smooth at the origin for $p \leq 1$, a pruning process is followed i.e., columns of matrix factors $\mathbf{U}$ and $\mathbf{V}$ whose energy is below a threshold are deleted. In doing so, division by zero is always avoided in the computation of $\mathbf{W}_{(\mathbf{U}^t, \mathbf{V}^t)}$.

Moreover, it can be easily shown that both $g(\mathbf{U}|\mathbf{U}^t, \mathbf{V}^t)$ and $q(\mathbf{V}|\mathbf{U}^{t+1}, \mathbf{V})$ are local tight upper-bounds of the original objective function since the difference of the approximate Hessian matrices from the true ones result to a positive semi-definite matrix (see [6] for details). Finally, convergence to stationary points of the original objective function can be established since all criteria required according to the BSUM framework, are satisfied, [4]. The outline of the algorithm is given in Algorithm 1. Note that the rank-one update strategy described in Section 4.2 (see Proposition 1) of the main paper is also adopted after the convergence of the algorithm to a stationary point for escaping potential poor local minima. The algorithm is assumed to converge when the relative error of successive reconstructions of matrix $\mathbf{X}$ defined as $\frac{\|\mathbf{U}^{t+1}\mathbf{V}^{t+1} - \mathbf{U}^t\mathbf{V}^t\|_F}{\|\mathbf{U}^t\mathbf{V}^t\|_F}$ becomes less than a threshold (denoted as tol in Algorithm 1) or the maximum iteration number is reached.

---

**Algorithm 1** The proposed Variational Schatten-$p$ matrix completion algorithm

---

Inputs: $P_Z(\mathbf{Y}), \lambda, \text{thres} = 10^{-5}, \text{MaxIter} = 1000, \text{tol} = 10^{-4}$
Initialize $\mathbf{U}^0 \in \mathbb{R}^{m \times d}, \mathbf{V}^0 \in \mathbb{R}^{n \times d}$
$t = 0$
**for** $t = 1 : MaxIter$ **do**
    $\mathbf{U}^{t+1} = \mathbf{U}^k - \nabla_\mathbf{U} \mathcal{L}(\mathbf{U}^t, \mathbf{V}^k)\tilde{\mathbf{H}}_{\mathbf{U}^t}^{-1}$
    $\mathbf{V}^{t+1} = \mathbf{V}^k - \nabla_\mathbf{V} \mathcal{L}(\mathbf{U}^{t+1}, \mathbf{V}^k)\tilde{\mathbf{H}}_{\mathbf{V}^t}^{-1}$
    Column Pruning:
    **if** $\|\mathbf{u}_i\|_2 \leq \text{thres}$ or $\|\mathbf{v}_i\|_2 \leq \text{thres}$, **then**
        Delete $\mathbf{u}_i$ and $\mathbf{v}_i$, $(i = 1, 2, \ldots, d)$.
    **end if**
    $t = t + 1$
    **if** $converged$ and rank-one update condition is satisfied **then**
        Add rank-one matrix for escaping poor local minima (see Proposition 1).
    **end if**
**end for**
Output: $\mathbf{U}^{t+1}, \mathbf{V}^{t+1}$

---