[Reviews · NeurIPS 2020]

Review 1

Summary and Contributions: This paper presents a new variational form of the Schatten-p quasi-norm via matrix factorization. The authors showed that the the local minimum of the factorization model is equivalent to the local minimum of the original model.

Strengths: The variational form of SVD-free Schatten-p quasi-norm appeared in previous work [4], in which the value of $p$ has discrete values. In the current work, the authors generalized the $p$ to arbitrary value in (0,1). The authors also provided some theoretical guarantees about the optimization and local minima.

Weaknesses: (1) The work is a little incremental. Although the authors extended the $p$ to arbitrary value, the impact is limited in practice because in previous work [4], for any $p$, one can always find a discrete p' such that p'<p. (2) The formulation (9) already appeared in the following paper: Giampouras et al. Alternating Iteratively Reweighted Least Squares Minimization for Low-Rank Matrix Factorization. IEEE TSP 2019. (3) The analysis of "escaping from poor local minima" is based on the rank one update. But the optimization algorithm for matrix completion is based on block successive minimization.

Correctness: Yes.

Clarity: Yes.

Relation to Prior Work: Yes.

Reproducibility: Yes

Additional Feedback: (1) The authors proposed to use rank one update to avoid poor local minima but in the experiment block successive minimization was utilized. The authors should explain the discrepancy. (2) Does Theorem 1 still hold for $p>1$? (3) It would be better if the authors could provide an intuitive example of the "poor" local minima in Section 4.2. %%% Thanks for the efforts. I am satisfied with the response. I increased the score to "accept"


Review 2

Summary and Contributions: This paper builds on a large literature of Schatten relaxations, and presents an elegant and compelling new approach to optimizing Schatten norms with p<1 without resorting to SVD computations. The authors also present an analysis showing the factorization does not introduce spurious local minima, and present a method to add a rank 1 component to reduce the rank further. Numerics validate the approach and show it to perform about as well as the best previous work.

Strengths: The factorization presented is more elegant than previous work while attaining similar performance. The theory presented is cogent and relevant: it is nice to know that the factorization does not introduce spurious local minima, and it is surely useful to know how to escape any local minima we may find.

Weaknesses: The method does not present an empirical improvement. The numerics are not so exciting. While the theory in this paper is stronger, it seems the method performs almost identically to the FGSR when the Schatten p norm target is similar.

Correctness: Yes, claims seem sound.

Clarity: Yes, it is easy to read, despite a few typos (noted below in detailed remarks).

Relation to Prior Work: Yes, the relation to previous work is clear. On the other hand, the idea in 4.2 is very far from new. For example, a similar idea was proposed in https://arxiv.org/abs/0807.4423. I wonder if these ideas might allow a generalization beyond RIP matrices?

Reproducibility: Yes

Additional Feedback: * line 22: typo, should be Frobenius * write out "with respect to" in text. * line 93: use an em-dash (written as three dashes --- in latex), not hyphen. * Theorem 3 Do \hat{U} and \hat{V} have to be in the specific form in theorem 3? Looks like it is necessary for \hat{U} and \hat{V} to be of the form of orthornomal matrix times positive diagonal matrix, which is not always satisfied for arbitrary local minima. Even though the paragraph after Theorem 3 says one can always achieve a local minima for X if a local minima for (U,V) is attained. Please explain. * What is R in Proposition 1? The delta = 0 case is quite rare and not so useful. Perhaps make it a remark instead of a part of the proposition. * Experiments seem to suggest smaller p is better; can you comment? The plots are also a bit difficult to see due to the many overlapping curves. * The author might want to provide a definition of regular subgradients in the appendix.


Review 3

Summary and Contributions: Authors propose to re-formulate the schatten p-norm (l_p norm of singular values of a matrix) using a variational form. They provide RIP based recovery results for matrix completion & propose to solve the problem using a conditional gradient a.k.a. Frank-Wolfe method

Strengths: Strength of the work are in discussing RIP-based recovery guarantees to S_p norms.

Weaknesses: The "novel" variational formulation is not novel. A more general statement is known since Jameson, Graham James Oscar. Summing and nuclear norms in Banach space theory. Vol. 8. Cambridge University Press, 1987, see also Convex relaxations of structured matrix factorizations https://arxiv.org/pdf/1309.3117.pdf Proposition 6 for example. Optimization using this norm hence becomes straightforward, see the arxiv reference above

Correctness: Looks correct to me

Clarity: Pretty clear

Relation to Prior Work: Missing references that I cited which question novelty of the work

Reproducibility: Yes

Additional Feedback:

[Author Response · NeurIPS 2020]

We thank the three reviewers (**R2,R3,R4**) for taking the time and effort to evaluate our paper and for the fruitful
comments they have made. We are also grateful to **R2,R3** for their positive evaluation of our paper and for finding our
work makes new, elegant, cogent and relevant contributions. Next, we address the reviewers' main critiques.

**[R2, (1)] The work is incremental. The impact is limited in practice compared to [4].** We disagree with the
reviewer. The proposed form supersedes the work in [4] in many aspects. Namely, [4] provided upper-bounds of the
Schatten-$p$ quasi-norm *only for specific discrete values of* $p$. In our work, we completely generalize the work of [4] by
providing variational forms for any continuous value of $p \in (0, 1]$, which we argue are also considerably simpler than
those presented in [4]. Given the formulation we are proposing we cannot envision a situation where the form in [4]
provides any advantages which would justify the disadvantage of only being able to use a discrete subset of $p$ values.

**[R2, (2)] The proposed variational form of Schatten-$p$ quasi-norm (9) already appeared in [C1].** This is not true.
In [C1], the authors use the same regularizer but no connection with the Schatten-$p$ quasi-norms defined on the product
space ($\mathbf{X}$) is provided. Precisely, the authors in [C1] miss the critical point that the regularizer is a tight upper bound of
the Schatten-$p$ quasi-norm, which is illustrated and proved in our paper.

**[R2, (3)] Rank one updates and the block successive minimization matrix completion algorithm.** The rank one
updating scheme is a "meta-algorithm" that is "activated" whenever the proposed block successive minimization based
matrix completion algorithm reaches a stationary point. We will clarify this point in the final version.

**[R3, (1)] The method does not present an empirical improvement. The numerics are not so exciting.** Presenting
empirical improvement as compared to the approach in [4] is not the objective of our paper. It comes as no surprise to
observe that the proposed approach performs comparably to the form in [4] since both forms are equivalent models w.r.t
$\mathbf{X}$ (assuming one selects a value of $p$ which can be used by the less-flexible form in [4]). The merits of our approach are
(a) a more general and simpler variational form of the Schatten-$p$ quasi-norm, (b) its favorable properties detailed in the
paper and (c) the rigorous theory provided for the analysis of the properties of local minima.

**[R4, (1)] The "novel" variational formulation is not novel. See [C2] and [C3].** It would appear as if the reviewer
may have missed the main point of the paper, as this is clearly not true. In our work, we deal with a setting which
is **nonconvex** even in the product space (i.e., nonconvex w.r.t. $\mathbf{X}$) arising from values of $p < 1$ in the Schatten-$p$
quasi-norm. Both [C2] and [C3] that are cited by the reviewer deal with **convex** scenarios, which are equivalent to the
variational form of the nuclear norm that we discuss in (2), and we note that establishing the extension of (2) to the
$p < 1$ form in (9) is non-trivial. For example, for $p > 1$ the form in (9) has no connection to a Schatten-$p$ function (see
the following answer).

Next we elaborate on additional feedback, comments and suggestions made by the reviewers.

**[R2] Is Theorem 1 valid for** $p > 1$**?** No. When $p > 1$ concavity no longer holds and the direction of the inequality
appearing in Theorem 1 reverses. In fact, for $p > 1$ it is simple to show that the optimization problems in (9) always
have an infimum of 0 regardless of the value of $\mathbf{X}$.

**[R2] Intuitive example of a "poor" local minima that we can escape from via the rank-one updating scheme.** An
intuitive example arises if we consider matrix factors with one pair of all-zero columns. It is easily shown the all-zero
columns will always be stationary, and the rank-one updating scheme allows us to escape from these "poor" minima.

**[R3] Rank-one updating scheme extension for non RIP matrices.** We thank the reviewer for bringing up this point,
as we are also interested in this question. The abstraction that allows the rank-one update to succeed is that for rank-one
perturbations the objective function can be reasonably bounded by the quadratic form in (43) plus some one-dimensional
function which depends only on the scale of the rank-one factors. We conjecture this same approach can be successfully
applied to many other problems, which we intend to explore in future research.

**[R3] On the properties of local minima in Theorem 3.** In Theorem 3 it is assumed that local minima satisfy certain
properties i.e., the matrix factors consist of pairwise orthogonal columns. It should be noted that in practice, we
can always reach a local minimizer of that form by performing an extra orthogonalization step to any arbitrary local
minimizer without those properties. Note, however, that we conjecture that these conditions are necessary for any local
minimizer and we are currently working on proving that this is a necessary condition of local minima for this problem.

**[R3] Improved empirical results for smaller** $p$**.** Indeed, it is empirically observed that as $p$ decreases towards 0, the
error also becomes lower. This is in line with the fact that Schatten-$p$ quasi-norm better approximates the rank function
for $p$ approaching 0. At some point one would expect the increased difficulty in optimization for small $p$ to offset the
advantages of the model for lower $p$, but we did not observe this in our experiments for the values of $p$ we explored.

[C1] Giampouras et al., *Alternating Iteratively Reweighted Least Squares Minimization for Low-Rank Matrix Factorization*, IEEE TSP, 2019.
[C2] Jameson, Graham James Oscar,*Summing and nuclear norms in Banach space theory.* Vol. 8. Cambridge University Press, 1987.
[C3] Francis Bach, *Convex relaxations of structured matrix factorizations*, 2013 (https://arxiv.org/pdf/1309.3117.pdf).


[Meta-Review · NeurIPS 2020]

The paper gives a novel SVD-free variational formulation of the Schatten-p matrix quasi-norm for p in (0,1), including an analysis of the problems caused by local minima and how to avoid them. The method is tested briefly on Schatten-norm matrix completion but the main emphasis of the paper is theoretical and the empirical improvements achieved relative to the directly competing FGSR method are limited. Two reviewers scored this above threshold, one below. In discussion the negative reviewer agreed that the method's ability to deal with p<1 is valuable, but still felt that the advance was too limited for acceptance. On balance the AC and SAC decided that the positive points were sufficient for acceptance given the importance of low-p Schatten norms for a range of ML methods. In the final paper, the discussion of the advantages of the new approach relative to FGSR needs to be strengthened.